# Adaptive Sample Sharing for Multi Agent Linear Bandits

Hamza Cherkaoui [1]   Merwan Barlier [1]   Igor Colin [2]

## Abstract

The multi-agent linear bandit setting is a well-known setting for which designing efficient collaboration between agents remains challenging. This paper studies the impact of data sharing among agents on regret minimization. Unlike most existing approaches, our contribution does not rely on any assumptions on the bandit parameters structure. Our main result formalizes the trade-off between the bias and uncertainty of the bandit parameter estimation for efficient collaboration. This result is the cornerstone of the Bandit Adaptive Sample Sharing (BASS) algorithm, whose efficiency over the current state-of-the-art is validated through both theoretical analysis and empirical evaluations on both synthetic and real-world datasets. Furthermore, we demonstrate that, when agents' parameters display a cluster structure, our algorithm accurately recovers them.

## 1. Introduction

The stochastic multi-armed bandit (MAB) framework provides efficient tools for solving sequential optimization problems (Lai & Robbins, 1985; Abbasi-Yadkori et al., 2011; Dani et al., 2008; Soare, 2015). In this setting, an agent repeatedly pulls an arm from a finite set of candidates and observes the associated noisy reward, sampled from an unknown fixed distribution.

A common extension of the MAB problem is the linear bandit setting (Lattimore & Szepesvári, 2020), where each arm is associated with a vector $x \in \mathbb{R}^d$ and its expected reward is a linear combination of the vector and an unknown bandit parameter $\theta^* \in \mathbb{R}^d$. The linear bandits framework is a generalization of MAB that takes into account the similarities between the arms to identify better pulling strategies. Efficiently solving this problem often consists in obtaining

an accurate estimate of the parameter $\theta^*$ with respect to some metric. For example, in the context of regret minimization (Li et al., 2010; Chu et al., 2011), the goal is to obtain an estimate of $\theta^*$ that is particularly accurate in the directions associated with large rewards.

In the last decade, there has been an increasing focus on the multi-agent setting (Cesa-Bianchi et al., 2013; Gentile et al., 2014; Nguyen & Lauw, 2014; Ban & He, 2021; Ghosh et al., 2022; Do et al., 2023; Yang et al., 2024), where multiple agents share samples to make better decisions together. For example, consider the problem of optimizing the parameters of wireless antennas in a network to improve the quality of service for nearby users. Antennas that belong to similar environments and constraints should yield similar rewards and are likely to benefit from sharing their observations, at least in the early stages of the optimization process. Indeed, if the rewards are derived from a similar parameter $\theta^*$, sharing observations effectively reduces the noise variance, making the estimator more accurate. Similar arguments can be made for recommendation systems or drug trials (Sarwar et al., 2002; Su & Khoshgoftaar, 2009; Li et al., 2010; Chu et al., 2011), where similar profiles should lead to similar observations.

This problem, known in the literature as Heterogeneous Multi-Agent Linear Stochastic Bandits, has been mostly approached using Euclidean similarity to define clusters of agents (Gentile et al., 2014; Nguyen & Lauw, 2014), within which sample sharing is allowed.

This paper develops a new approach to determine, *without assumption on a clustered structure*, when agents should share samples, focusing on the bias introduced by heterogeneous agents, the resulting reduction in uncertainty, and the trade-offs between these factors. Based on our similarity metric on the Mahalanobis distance induced by the observations to focus on *directions with large reward* and avoiding using the overly conservative Euclidean distance, we introduce the Bandit Adaptive Sample Sharing (BASS) algorithm. This method outperforms the state-of-the-art by pairing similarity learning with regret minimization.

We organize our paper as follows: first, in Section 2, we detail how this problem was addressed in the literature and summarize our contributions, Section 3 provides the intuition on how our approach challenges it. Then in Section 4,

---

[1]Huawei Noah's Ark Lab, France [2]LTCI, Télécom Paris, Institut Polytechnique de Paris, France. Correspondence to: Hamza Cherkaoui <hamza.cherkaoui@huawei.com>.

*Proceedings of the 42^{nd} International Conference on Machine Learning*, Vancouver, Canada. PMLR 267, 2025. Copyright 2025 by the author(s).

we formalize the linear bandit setting for the single agent case. Section 5 explores sample sharing in a multi-agent context and introduces our sharing criterion. In Section 6, we present and detail our algorithm, followed by a theoretical analysis. Finally, we empirically evaluate our method in Section 7.

## 2. Related Work

**Cluster structure known in advance** Some work has assumed that the parameter structure is known in advance. Cesa-Bianchi et al. (2013) modifies the problem variables to consider all agents simultaneously, allowing them to mimic a classical single-agent scenario while proposing an efficient multi-agent algorithm. In Wu et al. (2016), the problem is modeled to compute each reward as a linear combination of neighbors parameters, leading to a collaborative setting. More recently, Moradipari et al. (2022) defined a setting in which agents aim to maximize the average reward of the network, forcing agents to share their observations during a communication phase.

In real-world scenarios, however, the parameter structure is often unknown, so a natural goal is to recover it before enabling collaboration.

**Estimating the cluster structure as a graph** The seminal work of Gentile et al. (2014) introduces a novel collaborative framework: agents are grouped into clusters, and agents within the same cluster have the Euclidean distance between their bandit parameters smaller than a given threshold. With this assumption, the authors derive a similarity measure based on this distance and use the corresponding graph to recover clusters. Since then, variants have been proposed to improve the accuracy of the parameter estimates, *e.g.*, by averaging over connected components rather than neighborhoods (Li & Zhang, 2018), or by improving the robustness of the estimates by offsetting the OLS (Wang et al., 2023). Other work has focused on improving the quality of the clustering itself: Ban & He (2021) allows overlapping clusters, Li et al. (2019); Xiangyu et al. (2024) each proposes a splitting and merging procedure to correct for potential clustering errors, and Ghosh et al. (2022) aims to control the minimum cluster size. Although still based on a Euclidean similarity measure, Cheng et al. (2023) opts for a different clustering routine to take advantage of hedonic game theory. Do et al. (2023) and Yang et al. (2024) trigger collaboration rounds, controlled by a collaboration budget, when the design matrices deviate too much from each other. Finally, Li et al. (2016); Gentile et al. (2017) base their similarity on the rewards themselves, rather than the parameters, alleviating some of the dimensionality issues of the Euclidean similarity.

While most contributions have relied or focused on im-

proving the clustering process with the Euclidean similarity introduced by Gentile et al. (2014), this paper takes advantage of the Mahalanobis distance, as regret minimization problems require focusing primarily *on the directions with the highest rewards*. Furthermore, unlike most existing approaches that rely on the assumption of an *agent clustered structure*, our contribution does not impose such constraint.

**Multivariate approaches** In an opposite research direction, the approach in Nguyen & Lauw (2014) takes advantage of the $k$-means (MacQueen (1967)) algorithm to cluster the agents. Alternatively, instead of making the hypothesis of clusters, other approaches propose to enforce some regularization on the bandit parameter estimation to introduce structure between agents. The Non-negative Matrix Factorization (NMF) technique in Song et al. (2018) allows them to recover a probability distribution on the cluster labels for each agent. Finally, a low-rank decomposition of the stacked bandit parameter matrix in Yang et al. (2020) yields the desired cluster structure of the problem, by combining the decomposition with a sparsity constraint.

However, the multivariate aspect of these approaches makes them less suitable for a distributed framework, which is often required in real-world applications (*e.g.*, recommendation systems, wireless antenna network).

### 2.1. Our contributions

We propose a novel algorithm that addresses several previous limitations and introduces key features that distinguish it from previous work.

1. *No Assumption on Bandit Parameters*: Our approach does not rely on any assumptions on the bandit parameter structure to enable collaboration, and it learns to balance the trade-off between the collaborative bias and the uncertainty reduction.

2. *Anisotropic Approach*: In regret minimization, we focus on directions with large reward; our approach therefore focuses on the same directions to quantify the bias with adequate precision.

3. *Theoretical and Empirical Analysis*: We formally define the problem, conduct an exhaustive analysis of collaborative stopping time, and cumulative and instantaneous regret. In addition, we present a fair and comprehensive empirical evaluation of our algorithm using both synthetic and real-world data.

## 3. Preliminaries

### 3.1. Notation

We use lowercase (*e.g.*, $\alpha$) to denote a scalar, bold (*e.g.*, $\boldsymbol{x}$) to denote a vector, and uppercase bold (*e.g.*, $\boldsymbol{A}$) is reserved

for a matrix. The $\ell_2$-norm of a vector $\boldsymbol{x}$ is $\|\boldsymbol{x}\|_2 = \sqrt{\boldsymbol{x}^\top \boldsymbol{x}}$ and the Mahalanobis weighted $\ell_2$-seminorm is $\|\boldsymbol{x}\|_{\boldsymbol{A}} = \sqrt{\boldsymbol{x}^\top \boldsymbol{A} \boldsymbol{x}}$, where $\boldsymbol{A} \succeq 0$ is semidefinite positive. The ellipsoid of center $\boldsymbol{c} \in \mathbb{R}^d$, shape $\boldsymbol{A} \succeq 0$ and radius $r$ is denoted $\mathcal{E}(\boldsymbol{c}, \boldsymbol{A}, r)$, that is

$$\mathcal{E}(\boldsymbol{c}, \boldsymbol{A}, r) = \left\{ \boldsymbol{x} \in \mathbb{R}^d, \|\boldsymbol{x} - \boldsymbol{c}\|_{\boldsymbol{A}} \leq r \right\} \ .$$

An exhaustive notation summary can be found in the supplementary material Appendix B.

### 3.2. A simple two agent problem

To build intuition about sample sharing in the regret minimization problem, we first examine the simpler case of parameter estimation with two agents.

Consider two agents, each one with the aim of estimating its linear parameter $\boldsymbol{\theta}_i^* \in \mathbb{R}^d$, $i \in \{1, 2\}$. To do this, each agent $i$ has a series of $t$ actions $(\boldsymbol{x}_{i,1}, \ldots, \boldsymbol{x}_{i,t})$ associated with a series of $t$ noisy rewards $(y_{i,1}, \ldots, y_{i,t})$ such that for $1 \leq s \leq t$:

$$y_{i,s} = {\boldsymbol{\theta}_i^*}^\top \boldsymbol{x}_{i,s} + \eta_{i,s} \ ,$$

for some 1 sub-Gaussian[1] noise $\eta_{i,s}$. The unbiased Ordinary Least Squares (OLS) estimator is then:

$$\hat{\boldsymbol{\theta}}_{i,t} = \left( \sum_{s=1}^t \boldsymbol{x}_{i,s} \boldsymbol{x}_{i,s}^\top \right)^+ \sum_{s=1}^t y_{i,s} \boldsymbol{x}_{i,s} = \boldsymbol{\theta}_i^* + \boldsymbol{A}_{i,t}^+ \boldsymbol{z}_{i,t} \ ,$$

where $\boldsymbol{A}_{i,t} = \sum_{s=1}^t \boldsymbol{x}_{i,s} \boldsymbol{x}_{i,s}^\top$, $\boldsymbol{z}_{i,t} = \sum_{s=1}^t \eta_{i,s} \boldsymbol{x}_{i,s}$ and for any $\boldsymbol{M} \in \mathbb{R}^{d \times d}$, $\boldsymbol{M}^+$ is the Moore-Penrose inverse. For simplicity, we assume that both agents hold the same series of actions, so $\boldsymbol{A}_{1,t} = \boldsymbol{A}_{2,t} = \boldsymbol{A}_t$.

If the two agents shared their observations, the OLS of the merged dataset would be

$$\hat{\boldsymbol{\theta}}_{c,t} = \boldsymbol{\theta}_c^* + \frac{1}{2} \boldsymbol{A}_t^+ (\boldsymbol{z}_{1,t} + \boldsymbol{z}_{2,t}) \ ,$$

where $\boldsymbol{\theta}_c^* = \frac{1}{2}(\boldsymbol{\theta}_1^* + \boldsymbol{\theta}_2^*)$. If both parameters are equal, then the merged OLS improves each independent OLS, since it is unbiased and the noise is only $\frac{1}{\sqrt{2}}$ sub-Gaussian. However, if the parameters are different, the merged OLS is biased, and the question becomes whether the noise reduction outweighs the bias.

Since the overall goal is to consider the regret minimization problem, we measure the accuracy of an estimator by its Mahalanobis distance w.r.t. the design matrix $\boldsymbol{A}_t$. Indeed, the larger the reward, the more accurate the estimate needs to be. Using the analysis of Abbasi-Yadkori et al. (2011), we have, with high probability,

$$\|\hat{\boldsymbol{\theta}}_{i,t} - \boldsymbol{\theta}_i^*\|_{\boldsymbol{A}_t} \leq \beta_t \ ,$$

---

[1] see Assumption 4.2 for a formal definition of sub-Gaussianity.

where $\beta_t = O(\sqrt{\log(t)})$ will be detailed later. Similarly,

$$\|\hat{\boldsymbol{\theta}}_{c,t} - \boldsymbol{\theta}_i^*\|_{\boldsymbol{A}_t} \leq \frac{1}{2}\|\boldsymbol{\theta}_1^* - \boldsymbol{\theta}_2^*\|_{\boldsymbol{A}_t} + \frac{1}{\sqrt{2}}\beta_t \ .$$

Therefore, the collaborative estimate $\hat{\boldsymbol{\theta}}_{c,t}$ improves the regular OLS estimator when

$$\|\boldsymbol{\theta}_1^* - \boldsymbol{\theta}_2^*\|_{\boldsymbol{A}_t} \leq (2 - \sqrt{2})\beta_t \ . \tag{1}$$

Equation (1) suggests that collaboration reduces the error as long as the bias introduced $\|\boldsymbol{\theta}_1^* - \boldsymbol{\theta}_2^*\|_{\boldsymbol{A}_t}$ is less than a fraction of the uncertainty of the estimation $\beta_t$.

With this simple case, we argue that, given a sampling approach of the arms $(\boldsymbol{x}_s)_{s=1}^t$, we need to adapt *when the collaboration should occur* to accelerate the estimation process. The key point is to detect when the condition of Equation (1) stops being verified to end the collaboration and to avoid introducing a detrimental bias.

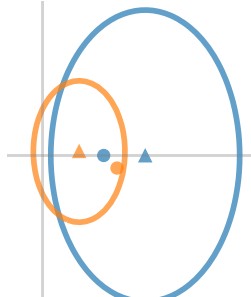

*Figure 1.* Estimation of $\boldsymbol{\theta}_1^*$ depicted with • (resp. $\boldsymbol{\theta}_c^*$ with •), $\hat{\boldsymbol{\theta}}_{1,t}$ with ▲ (resp. $\hat{\boldsymbol{\theta}}_{c,t}$ with ▲), along with the corresponding confidence ellipsoid in blue (resp. in orange). The collaborative estimate has a reduced uncertainty ellipsoid $\|\hat{\boldsymbol{\theta}}_{c,t} - \boldsymbol{\theta}_i^*\|_{\boldsymbol{A}_t}$.

The later intuitive result can be easily illustrated. In Figure 1, we display $\boldsymbol{\theta}_1^*$ and $\hat{\boldsymbol{\theta}}_t$ along with their confidence ellipsoid in blue and their collaborative counterparts in orange. We notice the well-known estimation bias - variance trade-off.

## 4. Single agent regret minimization

In the linear bandit setting, at each time step $t$ an agent chooses an arm $\boldsymbol{x}_t$ from a set $\mathcal{X} = (\boldsymbol{x}_k)_{k=1}^K \in \mathbb{R}^d$ and receives a reward signal $y_t = \boldsymbol{x}_t^\top \boldsymbol{\theta}^* + \eta$, where $\boldsymbol{\theta}^*$ is the (unknown) bandit parameter and $\eta$ is some centered noise. For the remainder of the paper, we assume that the arms and the bandit parameters are bounded as depicted in Assumption 4.1

**Assumption 4.1** (Bounded Norms). *We assume there exists a constant $L > 0$ such that the true parameter satisfies $\|\boldsymbol{\theta}^*\| \leq L$, and all context vectors satisfy $\|\boldsymbol{x}_k\| \leq 1$ for all $1 \leq k \leq K$.*

Let $\mathcal{F}_t$ be the $\sigma$-algebra generated by $(y_1, \ldots, y_{t-1}, \boldsymbol{x}_1, \ldots, \boldsymbol{x}_{t-1}, \boldsymbol{x}_t)$, then $\boldsymbol{x}_t$ is $\mathcal{F}_t$-measurable and $y_{t-1}$ is $\mathcal{F}_t$-measurable. Moreover, we make the common Assumption 4.2 that the noise is conditionally $R$-sub-Gaussian for some constant $R > 0$.

**Assumption 4.2** ($R$-sub-Gaussianity of the Noise)**.** *The noise term $\eta$ is conditionally $R$-sub-Gaussian for some constant $R > 0$. That is, for all $\lambda \in \mathbb{R}$,*

$$\mathbb{E}\left[\exp(\lambda \eta) \mid \mathcal{F}_{t-1}\right] \leq \exp\left(\frac{\lambda^2 R^2}{2}\right) \ .$$

The goal is to minimize the cumulative pseudo-regret, formally defined in Audibert et al. (2009) as $R_T = \sum_{t=1}^{T} r_t$ , with $r_t = \boldsymbol{\theta}^{*\top}(\boldsymbol{x}^* - \boldsymbol{x}_t)$ and $\boldsymbol{x}^* = \arg\max_k \boldsymbol{\theta}^{*\top} \boldsymbol{x}_k$. For this purpose, the OFUL algorithm (Abbasi-Yadkori et al., 2011) implements the principle of optimism in the face of uncertainty in the linear bandit case. At each time step $t$, the learner pulls the arm that maximizes the expected reward associated with the best parameter in a confidence region around the Ordinary Least Squares (OLS) estimator. Formally, at time $t + 1$, given a previous OLS estimator $\hat{\boldsymbol{\theta}}_t$ and a $\mathcal{C}_\delta(\hat{\boldsymbol{\theta}}_t)$, the selected arm maximizes the optimist reward

$$\boldsymbol{x}_{t+1} \in \arg\max_{\boldsymbol{x} \in \mathcal{X}} \max_{\boldsymbol{\theta} \in \mathcal{C}_\delta(\hat{\boldsymbol{\theta}}_t)} \boldsymbol{x}^\top \boldsymbol{\theta} \ ,$$

where $\mathcal{C}_\delta(\hat{\boldsymbol{\theta}}_t)$ is such that $\mathbb{P}(\boldsymbol{\theta}^* \in \mathcal{C}_\delta(\hat{\boldsymbol{\theta}}_t)) \geq 1 - \delta$. In practice, the confidence ellipsoid $\mathcal{C}_\delta(\hat{\boldsymbol{\theta}}_t)$ is constructed from the previous pulls using Hoeffding's concentration inequality (Tropp et al., 2015), which leads to the theorem Theorem 4.1 from Abbasi-Yadkori et al. (2011):

**Theorem 4.1** (Confidence Ellipsoid for Bandit Parameter Estimation)**.** *Let $\delta \in (0, 1)$, $t > 0$, and $\boldsymbol{\theta}^* \in \mathbb{R}^d$. Let $(\boldsymbol{x}_s)_{1 \leq s \leq t}$ denote the sequence of arms pulled up to time $t$. Under Assumption 4.1 and Assumption 4.2, the following holds with probability at least $1 - \delta$:*

$$\mathbb{P}\left(\boldsymbol{\theta}^* \in \mathcal{C}_\delta(\hat{\boldsymbol{\theta}}_t)\right) \geq 1 - \delta \ ,$$

*where*

$$\begin{cases} \mathcal{C}_\delta(\hat{\boldsymbol{\theta}}_t) = \left\{ \left\|\hat{\boldsymbol{\theta}}_t - \boldsymbol{\theta}^*\right\|_{\boldsymbol{A}_t} \leq \beta(\delta, \boldsymbol{A}_t) \right\} \ , \\ \beta(\delta, \boldsymbol{A}_t) = R\sqrt{2\log\left(\frac{1}{\delta}\sqrt{\frac{\det(\boldsymbol{A}_t)}{\det(\boldsymbol{A}_0)}}\right)} \ , \end{cases} \quad (2)$$

*and $\boldsymbol{A}_t = \sum_{s=1}^{t} \boldsymbol{x}_s \boldsymbol{x}_s^\top$ is the empirical design matrix.*

For the remainder of the paper, we assume that $\mathcal{X}$ spans $\mathbb{R}^d$. Also, we set $\boldsymbol{A}_0 = \boldsymbol{I}$, so:

$$\beta(\delta, \boldsymbol{A}_t) = R\sqrt{2\log\frac{1}{\delta} + \log\det(\boldsymbol{A}_t)} \ .$$

# 5. Adaptive sample sharing

In this section, we consider the collaborative setting and examine the impact of sharing samples on regret minimization.

## 5.1. Collaborative setting

We consider a multi-agent setting where each agent $i \in \{1, \ldots, N\}$ is able to pull arms from a linear bandit of unknown parameters $\boldsymbol{\theta}_i^* \in \mathbb{R}^d$. Every linear bandit uses the same set of arms $\mathcal{X}$. We assume that the historical data are known to a central controller. At each time step $t$, the controller selects the arm to be pulled by the agents and observes the noisy rewards.

We are interested in comparing two pulling strategies: a local strategy, where each agent can only use its own observations to derive a pulling policy, and a collaborative strategy, where agents are allowed to regroup their observations in order to derive more general policies. Throughout the paper, we denote by $\widehat{\mathcal{N}}_i(t)$ the set of agents sharing their data with agent $i$ at time $t$, we index a quantity by $i$ if it is computed using only local observations of agent $i$, and by $\widehat{\mathcal{N}}_i(t)$ if it is computed using observations from the set of all agents that share their data with $i$ at time $t$. In both cases, we focus on an OFUL approach, so the two strategies can be summarized as follows.

**Local strategy:** use OFUL with local observations, compute the local OLS $\hat{\boldsymbol{\theta}}_i$ and the confidence region $\mathcal{C}_\delta(\hat{\boldsymbol{\theta}}_i)$;

**Collaborative strategy:** use OFUL with the observations of the collaborating agents, compute the associated OLS $\hat{\boldsymbol{\theta}}_{\widehat{\mathcal{N}}_i(t)}$ and the confidence region $\mathcal{C}_\delta(\hat{\boldsymbol{\theta}}_{\widehat{\mathcal{N}}_i(t)})$.

## 5.2. Separation test

We now analyze the impact of sample sharing on the regret and formally introduce the test to detect when beneficial sharing stops. The arm selection strategy is based on OFUL and depends on whether or not the observations are shared to define the OLS estimator and the confidence ellipsoid. Unlike the analysis in Subsection 3.2, we focus on regret minimization rather than parameter estimation. To this end, we recall the following bounds on OFUL instantaneous regret for both independent and data-sharing settings.

**Lemma 5.1** (Instantaneous Regret Upper Bounds)**.** *Let $0 < \delta < 1$ and $t > 0$. Let $(\boldsymbol{\theta}_j^*)_{1 \leq j \leq N} \in \mathbb{R}^{d \times N}$ be the true parameters of $N$ linear bandit models. Denote by $\boldsymbol{x}_{i,t}$ (resp. $\boldsymbol{x}_{\widehat{\mathcal{N}}_i(t)}$) the arm selected by agent $i$ using the local (resp. collaborative) strategy. Then, with probability at least $1 - \delta$, the instantaneous regret of agent $i$ is bounded as follows:*

*(i) Local strategy:*

$$r_i(t) \leq 2\beta\left(\delta, \boldsymbol{A}_{i,t}\right) \|\boldsymbol{x}_{i,t}\|_{\boldsymbol{A}_{i,t}^{-1}} \ .$$

*(ii) Collaborative strategy:*

$$r_i(t) \leq \frac{2\beta(\delta, m\boldsymbol{A}_{i,t})}{\sqrt{m}} \|\boldsymbol{x}_{\widehat{\mathcal{N}}_i(t)}\|_{\boldsymbol{A}_{i,t}^{-1}} + \frac{2}{m}\sum_{j \in \widehat{\mathcal{N}}_i(t)} \Delta_{\boldsymbol{A}_{i,t}}^{i,j} \ ,$$

*where* $m = |\widehat{\mathcal{N}}_i(t)|$ *and* $\Delta_{\boldsymbol{A}_{i,t}}^{i,j} = \|\boldsymbol{\theta}_i^* - \boldsymbol{\theta}_j^*\|_{\boldsymbol{A}_{i,t}}$.

Following the analysis of Subsection 3.2 and substituting the unknown parameters in their OLS estimates, we compare the right-hand side of the inequalities. For $j \in \widehat{\mathcal{N}}_i(t)$, we outline the same condition than Equation (1), see Appendix G:

$$\|\hat{\boldsymbol{\theta}}_i - \hat{\boldsymbol{\theta}}_j\|_{\boldsymbol{A}_{i,t}} \leq (2 - \sqrt{2})\beta(\delta, \boldsymbol{A}_t) \ . \tag{3}$$

We almost recover the overlapping ellipsoid test derived in Gilitschenski & Hanebeck (2012) and based on the following function:

$$\kappa(i,j) = 1 - \min_{s \in ]0,1[} \|\hat{\boldsymbol{\theta}}_i - \hat{\boldsymbol{\theta}}_j\|_{\left(\frac{\beta_i}{1-s}\boldsymbol{A}_{i,t}^{-1} + \frac{\beta_k}{s}\boldsymbol{A}_{j,t}^{-1}\right)^{-1}} \ . \tag{4}$$

The ellipsoid separation property is obtained from the sign of $\kappa(i,j)$, that is $\mathcal{E}(\hat{\boldsymbol{\theta}}_i, \boldsymbol{A}_{i,t}, \beta_i) \cap \mathcal{E}(\hat{\boldsymbol{\theta}}_j, \boldsymbol{A}_{j,t}, \beta_j) = \emptyset$ if and only if $\kappa(i,j) \leq 0$.

Since we consider a synchronous pulling, we have $\boldsymbol{A}_{i,t} = \boldsymbol{A}_{j,t} = \boldsymbol{A}_t$, and $\beta_i = \beta_j = \beta(\delta, \boldsymbol{A}_t)$. Indeed, in this setting, all collaborating agents select an arm based on the same shared history. Choosing a deterministic arm policy yields that two agents using the same history will select the same arm. Therefore, their design matrices will also be identical. The separation condition can be reformulated as stated in the following corollary.

Since we consider a synchronous setting, we have $\boldsymbol{A}_{i,t} = \boldsymbol{A}_{j,t} = \boldsymbol{A}_t$ and $\beta_i = \beta_j = \beta(\delta, \boldsymbol{A}_t)$. In this case, all collaborating agents operate on the same shared history. Because the arm selection policy (*e.g.*, UCB) is deterministic, agents with identical histories will always select the same arms. As a result, their design matrices evolve identically over time.

Under this assumption, the separation condition can be reformulated as stated in the following corollary.

**Lemma 5.2** (Ellipsoid Separation Under Synchronous Pulling). *Let* $0 < \delta < 1$, *and consider two agents* $i$ *and* $j$. *Under* Assumption 4.1 *and* Assumption 4.2, *define* $\tilde{\beta} = (1 + 1/\sqrt{2})\beta(\delta, \boldsymbol{A}_t)$. *Then the following statements are equivalent:*

*(i)* $\left\|\hat{\boldsymbol{\theta}}_i - \hat{\boldsymbol{\theta}}_j\right\|_{\boldsymbol{A}_t} \geq (2 + \sqrt{2})\beta(\delta, \boldsymbol{A}_t) \ ,$

*(ii)* $\mathcal{E}(\hat{\boldsymbol{\theta}}_i, \boldsymbol{A}_t, \tilde{\beta}) \cap \mathcal{E}(\hat{\boldsymbol{\theta}}_j, \boldsymbol{A}_t, \tilde{\beta}) = \emptyset \ .$

This shed a new light on the preliminary work in Subsection 3.2 and Subsection 5.2. The collaborative condition in Equation (1) matches the separation condition for ellipsoids $\mathcal{E}(\hat{\boldsymbol{\theta}}_i, \boldsymbol{A}_t, \tilde{\beta})$ and $\mathcal{E}(\hat{\boldsymbol{\theta}}_j, \boldsymbol{A}_t, \tilde{\beta})$. Interestingly, denoting as $\tilde{\delta}$ the value for which $\tilde{\beta} = \beta(\tilde{\delta}, \boldsymbol{A}_t)$, the bias is detrimental when we know with probability at least $1 - \tilde{\delta}$ that both

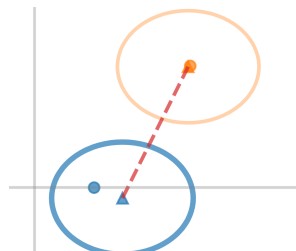

*Figure 2.* Estimation of $\boldsymbol{\theta}_i^*$ depicted with • (resp. $\boldsymbol{\theta}_j^*$ with •), $\hat{\boldsymbol{\theta}}_{i,t}$ with ▲ (resp. $\hat{\boldsymbol{\theta}}_{j,t}$ with ▲), along with the corresponding confidence ellipsoid in blue (resp. in orange). With high probability, the bias, depicted with a red dashed line - - - -, becomes detrimental when the ellipsoids are separated.

agents follow two different bandit parameters, as illustrated in Figure 2.

It should be noted that collaboration criteria based on the Euclidean distance (*e.g.*, Gentile et al. (2014); Ban & He (2021); Wang et al. (2023)) are equivalent to the Gilitschenski & Hanebeck (2012) test where the ellipsoids considered are $\mathcal{E}(\hat{\boldsymbol{\theta}}_i, \lambda_{\min}(\boldsymbol{A}_t)\boldsymbol{I}, \tilde{\beta})$ and $\mathcal{E}(\hat{\boldsymbol{\theta}}_j, \lambda_{\min}(\boldsymbol{A}_t)\boldsymbol{I}, \tilde{\beta})$, *ignoring most of the arm pulling history.*

In practice, computing (4) is expensive, instead we consider a cheaper, equivalent test function[2], defined as follows.

**Definition 5.1** ($\gamma$-relaxed ellipsoid separation test function). *For any iteration* $t$ *and any two agents* $i$ *and* $j$, *we denote as* $\Psi(i,j,t)$ *the quantity*

$$\Psi(i,j,t) = \mathbb{1}\left\{\min_{s \in [0,1]} \tau_t^{i,j}(s) < 0\right\} \ ,$$

*with*

$$\tau_t^{i,j}(s) = \frac{\gamma^2}{4} - \sum_{l=1}^{d} \mu_{t,l}^2 \cdot \frac{s(1-s)}{\beta_{i,t}^2 + s\left(\beta_{j,t}^2 \eta_{t,l} - \beta_{i,t}^2\right)}$$

*where* $\gamma > 0$ *and* $\boldsymbol{\mu} = \boldsymbol{\Phi}^\top(\hat{\boldsymbol{\theta}}_{i,t} - \hat{\boldsymbol{\theta}}_{j,t})$ *such as* $\boldsymbol{\eta}$ *and* $\boldsymbol{\Phi}$ *are the eigenvalues and the eigenvectors of the generalized eigenvalue problem:*

$$\boldsymbol{A}_{i,t}\boldsymbol{\Phi} = \boldsymbol{A}_{j,t}\boldsymbol{\Phi}\text{diag}(\boldsymbol{\eta}) \quad \text{with} \ \boldsymbol{\Phi}_i^\top \boldsymbol{A}_{j,t}\boldsymbol{\Phi}_i = 1 \ .$$

The parameter $\gamma$ in $\Psi$ is related to the ellipsoid radius and therefore depends on the level of confidence $\tilde{\delta}$ required for the separation.

**5.3. Separation time**

Two agents $i$ and $j$ may have a *beneficial collaboration* as long as $\Psi(i,j,t) = 1$. Therefore, we consider collaboration sets of the form $\widehat{\mathcal{N}}_{i,t} = \{j \mid \Psi(i,j,t) = 1\}$, and define the separation time between two agents as follows. Note that with our algorithm, once two agents are separated, they no longer share samples.

---

[2]We proof the ellipsoid separation property of $\Psi$ in the supplementary material Appendix I

**Definition 5.2** (Separation Time). *For two agents $i$ and $j$, the* separation time $T_s(i,j)$ *is defined as*

$$T_s(i,j) = \min\{t \in \mathbb{N} \mid \Psi(i,j,t) = 0\} \ ,$$

The following result gives a lower bound on the separation time based on the parameter gap.

**Theorem 5.1** (Lower Bound on the Separation Time $T_s$ Between Two Agents). *Let $0 < \delta < 1$. Consider two agents $i$ and $j$, and suppose that Assumption 4.1 and Assumption 4.2 hold. Then, with probability at least $1 - \delta$, the separation time satisfies:*

$$T_s(i,j) \geq \left\lceil \frac{8\left(\mathrm{erf}^{-1}(1-\delta) - \mathrm{erf}(\delta)\right)^2}{\|\boldsymbol{\theta}_i^* - \boldsymbol{\theta}_j^*\|_2} \right\rceil \ ,$$

*where $\mathrm{erf}(\cdot)$ denotes the Gauss error function.*

# 6. The Bandit Adaptive Sample Sharing algorithm

## 6.1. Description of the algorithm

Following Definition 5.1, we define a similarity graph $\mathcal{G}_t = (V, E_t)$, where $V = \{1, \ldots, N\}$ and $E_t = \{i, j \mid \Psi(i,j,t) = 1\}$, initialized as a complete graph. Therefore, agents $i$ and $j$ share their observations if and only if $(i,j) \in E_t$. The Bandit Adaptive Sample Sharing (BASS) Algorithm goes as follows.

At each iteration $t$, for each agent $i \in V$[3], the shared estimator $\hat{\boldsymbol{\theta}}_{\mathcal{N}_i(t)}$ is updated and an arm is pulled according to OFUL policy with an exploration parameter $\alpha$. The agent then uses the observed rewards to update its local estimate $\hat{\boldsymbol{\theta}}_i$. Once every active agent has pulled an arm and updated its local estimate, the similarity graph $\mathcal{G}_t$ is updated according to Definition 5.1. The complete procedure is detailed in Algorithm 1.

The overall complexity of the *BASS* algorithm is $\mathcal{O}\left(T(N + Kd^2 + (N+1)d^3)\right)$. We report the computational details and the associated complexity in the supplementary material Appendix Q.

Moreover, let $C(T)$ denote the total communication cost up to time $T$. In the worst-case scenario—where all $N$ agents share identical bandit parameters and communicate their observations at each round—we have $C(T) \leq (N-1)T$.

## 6.2. Theoretical analysis

**Separation time upper-bound** To facilitate this theoretical analysis, we consider a slightly modified version of

---

[3]See the experiments in Section 7 for the case of a single agent pulling an arm at each iteration.

---

**Algorithm 1** Bandit Adaptive Sample Sharing (*BASS*) algorithm

**Input**:
The UCB parameter $\alpha$, the exploration parameter $\epsilon_e$ and the confidence level $\delta$.

**Initialization**:
For $i = 1, \ldots, N$ set $\boldsymbol{b}_{i,0} = \boldsymbol{0}_d$, $\boldsymbol{A}_{i,0} = \boldsymbol{I}_d$
and the *complete* graph $\mathcal{G}_0 = (V, E_0)$ ;

1: **for** $t = 1, \ldots, T$ **do**
2:   **for** $i \in V$ **do**
3:     Determine the neighbors $\widehat{\mathcal{N}}_{i,t-1}$ of agent $i$ ;
4:     Update its current neighbor weights, set:
$$\boldsymbol{A}_{\widehat{\mathcal{N}}_{i_t,t-1}} = \boldsymbol{A}_{i,0} + \sum_{j \in \hat{\mathcal{N}}_{i,t-1}} (\boldsymbol{A}_{j,t-1} - \boldsymbol{A}_{j,0}) ,$$
$$\boldsymbol{b}_{\widehat{\mathcal{N}}_{i_t,t-1}} = \sum_{j \in \widehat{\mathcal{N}}_{i_t,t-1}} \boldsymbol{b}_{j,t-1} ,$$
$$\widehat{\boldsymbol{\theta}}_{\widehat{\mathcal{N}}_{i_t,t-1}} = \boldsymbol{A}_{\widehat{\mathcal{N}}_{i_t,t-1}}^{-1} \boldsymbol{b}_{\widehat{\mathcal{N}}_{i_t,t-1}} ;$$
5:     Select $k_t \in \underset{k=1,\ldots,K}{\arg\max} \ \widetilde{\boldsymbol{\theta}}_{\widehat{\mathcal{N}}_{i_t,t-1}}^{\top} \boldsymbol{x}_k + \beta_{\boldsymbol{A}_{\widehat{\mathcal{N}}_{i_t,t-1}}^{-1}}(\boldsymbol{x}_k)$
    with $\beta_{\boldsymbol{A}_{\widehat{\mathcal{N}}_{i_t,t-1}}^{-1}}(\boldsymbol{x}) = \alpha\sqrt{\boldsymbol{x}^{\top} \boldsymbol{A}_{\widehat{\mathcal{N}}_{i_t,t-1}}^{-1} \boldsymbol{x} \log(t)}$
6:     Pull $k_t$ and observe payoff $y_t$ ;
7:     Update weights, set:
$$\boldsymbol{A}_{i,t} = \boldsymbol{A}_{i,t-1} + \boldsymbol{x}_{k_t} \boldsymbol{x}_{k_t}^{\top},$$
$$\boldsymbol{b}_{i,t} = \boldsymbol{b}_{i,t-1} + y_t \boldsymbol{x}_{k_t} ;$$
8:   **end for**
9:   Update graph $\mathcal{G}_t = (N, E_t)$:
    Remove all $(i,j)$ such as $\Psi(i,j,t) = 0$ ;
10: **end for**

---

the algorithm *BASS* where the OFUL sampling is replaced with a uniform sampling with probability $\epsilon_e$, to ensure exploration in all directions. Empirically, we did not notice any difference with the above version.

**Theorem 6.1** (Upper Bound on the Separation Time $T_s$). *Let $0 < \delta < 1$, and consider two agents $i$ and $j$. Let $\epsilon_e > 0$ satisfy*

$$\epsilon_e \geq \frac{(1 + \sqrt{2})^2 R^2}{Q_{i,j}}, \ \text{with} \ Q_{i,j} = \frac{1}{K} \sum_{k=1}^{K} \left(\boldsymbol{x}_k^{\top}(\boldsymbol{\theta}_i^* - \boldsymbol{\theta}_j^*)\right)^2 \ .$$

*Then, under Assumption 4.1, Assumption 4.2, and following Figure 1, we have, with probability at least $1 - \delta$,*

$$T_s(i,j) \leq \left\lceil \frac{4(2 + \sqrt{2})^2 R^2 \log \frac{1}{\delta}}{2\epsilon_e Q_{i,j} - (2 + \sqrt{2})^2 R^2} \right\rceil \ .$$

**Cumulative pseudo-regret upper bound** We focus on the particular case of collaboration between two agents, as it can be easily generalized to $N$ agents by considering all pairs. We focus on the cumulative regret at separation time, as the problem transitions to the well-known single agent setting afterward: $R_{i,T}^{\text{collab}} = R_{i,0,T_s(i,j)}^{\text{collab}} + \sum_{t=T_s(i,j)}^{T} r_{i,t}$ , with $R_{i,0,T_s(i,j)}^{\text{collab}} = \sum_{t=0}^{T_s(i,j)} r_{i,t}$.

**Theorem 6.2** (Individual Regret During the Collaboration Phase). *Let $T_s = T_s(i,j)$ denote the separation time introduced in Definition 5.2. Under Assumption 4.1 and Assumption 4.2, the cumulative regret of agent $i$ during the collaboration phase is bounded as:*

$$R_{i,0,T_s}^{\text{collab}} \leq \mu(\delta, d, \gamma) \cdot \nu(\delta, d, T_s) + \frac{T_s}{2} \left( \boldsymbol{\theta}_i^* - \boldsymbol{\theta}_j^* \right)^\top \boldsymbol{x}_i^* \ ,$$

*where*

$$\mu(\delta, d, \gamma) = \frac{1}{2} + \frac{\gamma}{4} + \frac{1}{2\sqrt{2}} \sqrt{1 + \frac{d \log 2}{2 \log \frac{1}{\delta}}} \ ,$$

*and*

$$\nu(\delta, d, T_s) = \sqrt{4\beta(\delta, \boldsymbol{A}_{T_s})^2 T_s d \log \left( 1 + \frac{T_s}{d} \right)} \ .$$

With $\nu(\delta, d, T_s)$ we recognize the upper bound in the case of a single agent as derived in Abbasi-Yadkori et al. (2011), thus $\mu(\delta, d, \gamma)$ represents the collaborative gain and $\frac{T_s}{2}(\boldsymbol{\theta}_i^* - \boldsymbol{\theta}_j^*)^\top \boldsymbol{x}_i^*$ the result of the induced bias. Setting $\gamma < 2 - \sqrt{2}\sqrt{1 + (d \log 2)/(2 \log \frac{1}{\delta})}$, we have $\mu(\delta, d, \gamma) < 1$.

**Network cumulative pseudo-regret upper bound** Looking at the cumulative regret of all agents, we can further refine our upper bound. Let us set $\bar{R}_{0,T_s}^{\text{collab}} = \frac{1}{2} \sum_{i=1}^2 R_{i,0,T_s}^{\text{collab}}$, the averaged cumulative pseudo-regret.

**Theorem 6.3** (Regret During the Collaboration Phase). *Let $T_s = T_s(i,j)$ denote the separation time introduced in Definition 5.2. Under Assumption 4.1 and Assumption 4.2, the cumulative regret of all agents during the collaboration phase is bounded as:*

$$\bar{R}_{0,T_s}^{\text{collab}} \leq \mu'(\delta, d) \cdot \nu(\delta, d, T_s) + \frac{T_s}{4} \left( \boldsymbol{\theta}_i^* - \boldsymbol{\theta}_j^* \right)^\top \left( \boldsymbol{x}_i^* - \boldsymbol{x}_j^* \right) \ ,$$

*where*

$$\mu'(\delta, d) = \frac{1}{\sqrt{2}} \sqrt{1 + \frac{d \log 2}{2 \log \frac{1}{\delta}}} \ ,$$

*and $\nu(\delta, d, T_s)$ is as previously defined.*

Similarly to Theorem 6.2 we recognize the upper bound in the case of a single agent, thus $\mu'(\delta, d)$ represents the collaborative gain and $\frac{T_s}{4}(\boldsymbol{\theta}_i^* - \boldsymbol{\theta}_j^*)^\top(\boldsymbol{x}_i^* - \boldsymbol{x}_j^*)$ bias-related quantity.

**Regret analysis with clustered parameters** Most existing algorithms rely on the assumption that the linear bandit parameters are clustered around a few centroids. We thus extend our analysis to this specific assumption, which is formalized as:

**Assumption 6.1** (Clustering Structure). *We assume that there exist $M \ll N$ distinct bandit parameters $\{\boldsymbol{\theta}_1^*, \ldots, \boldsymbol{\theta}_M^*\} \subset \mathbb{R}^d$ such that, for all agents $i \in \{1, \ldots, N\}$, the true parameter satisfies:*

$$\boldsymbol{\theta}_i^* \in \{\boldsymbol{\theta}_1^*, \ldots, \boldsymbol{\theta}_M^*\} \ .$$

Under the assumption of clustered parameters, the collaboration may *always be beneficial*. We also introduce additional notation: for a given agent $i$, we define its true neighborhood (*i.e.*, the set of agents within its cluster) as $\mathcal{N}_i$. We also introduce the cardinalities $N_i = \rho_{m(i)} N = |\mathcal{N}_i|$ and $\hat{N}_{i,t} = |\widehat{\mathcal{N}_{i,t}}|$. Finally, we denote by $N_{i,t}^e$ the gap between both cardinalities, that is, $N_{i,t}^e = |\hat{N}_{i,t} - N_{i,t}|$. The following result upper bounds the number of misassigned agents.

**Lemma 6.1** (Expected Number of Misassigned Agents). *Let $0 < \delta < 1$. Under Assumption 4.1, Assumption 4.2, and Assumption 6.1, we have, with probability at least $1 - \delta$:*

$$\mathbb{E}\left[ N_i^e(t) \right] \leq n_i^e(t) \ ,$$

*where*

$$n_i^e(t) = \left\lceil \frac{N \sqrt{\det(\boldsymbol{A}_t)}}{\exp \left( \frac{1}{2} \left( \Delta_{\boldsymbol{A}_t}^{\min} - \gamma\beta(\delta, \boldsymbol{A}_t) \right)^2 \right)} - \rho_{m(i)} N \right\rceil \ ,$$

*and*

$$\Delta_{\boldsymbol{A}_t}^{\min} = \min_{i \neq j} \|\boldsymbol{\theta}_i^* - \boldsymbol{\theta}_j^*\|_{\boldsymbol{A}_t} \ .$$

With this clustering error bound, we can derive an upper bound on the cumulative pseudo-regret denoted $R_{i,0,T}^{\text{cluster}}$. We provide additional theoretical results in the supplementary material Appendix P.

**Theorem 6.4** (Cumulative Pseudo-Regret with Clustered Agents). *Under Assumption 4.1, Assumption 4.2, and Assumption 6.1, after $T$ iterations of the BASS algorithm, the cumulative pseudo-regret of agent $i$ satisfies:*

$$R_{i,0,T}^{\text{cluster}} \leq \mu''(\delta, d, N) \cdot \nu(\delta, d, T) + \frac{4L}{\delta \rho_{\min} N} \cdot C_i(T_c) \ ,$$

*where:*

$$C_i(T_c) = \sum_{t=1}^{\min(T, T_c)} n_i^e(t) \ ,$$

$$\mu''(\delta, d, N) = \frac{1}{\sqrt{\rho_{\min} N}} \sqrt{1 + \frac{d \log N}{2 \log \frac{1}{\delta}}} \ ,$$

$$T_c = \max_j T_s(i,j), \quad \rho_{\min} = \min_j \rho_{m(j)} \ ,$$

*and $\nu(\delta, d, T)$ is as previously defined.*

**Discussion** To better interpret this result, we derive a lower bound on the cumulative pseudo-regret in the idealized case where all agents share the same bandit parameter and form a single, known cluster, we obtain:

$$R_{i,0,T}^{\text{cluster}} \geq \frac{\nu(\delta, d, T)}{N} \quad .$$

This bound emphasizes the additional cost term $\frac{4L}{\delta \rho_{\min} N} \cdot C_i(T_c)$ in Theorem 6.4, which accounts for the effort required to correctly identify agent clusters.

Additionally, we summarize in Table 1 the upper bounds available for the concurrent algorithms considered under the structure assumption of the clustered agents. We provide the derivation steps for this table and include an additional oracle case in the supplementary materials Appendix P.

|  | **Cumululative pseudo regret** | **Expected number of misassigned agents** |
|---|---|---|
| DynUCB | not available | not available |
| CLUB | $\mathcal{O}\left(d\sqrt{\frac{M}{N}}\sqrt{T}\log(T)\right)$ | not available |
| SCLUB | $\mathcal{O}\left(d\sqrt{\frac{M}{N}}\sqrt{T}\log(T)\right)$ | not available |
| CMLB | $\mathcal{O}\left(\frac{\sqrt{d}}{\sqrt{\rho^{\min}N}}\sqrt{T}\log(T)^2\right)$ | not available |
| **BASS** | $\mathcal{O}\left(\frac{\sqrt{d}}{\sqrt{\rho^{\min}N}}\sqrt{T}\log(T/d)\right)$ | $\mathcal{O}\left(\frac{N}{\sqrt{T}^{\gamma^2 R^2 - 1}}\right)$ |

*Table 1.* Summary of available upper-bounds for clustered agents.

Our analysis of the cumulative pseudo-regret with clustered agents establishes the tightest upper bounds, incorporating a $\sqrt{d}/\sqrt{\rho^{\min}N}$ constant term and a $T/d$ term within the logarithm. Furthermore, to our knowledge, this is the first work to provide an analysis of the clustering error of the algorithm.

# 7. Experiment

All upper bounds include an *gain* term compared to the single agent case and a *loss* term from the introduced bias. In this section, we demonstrate that the combination of these two terms results in an overall *improved regret minimization* on both synthetic and real data experiments. The experiments were run in Python on 50 '*Intel Xeon @ 3.20 GHz*' CPUs and lasted a week. The code is publicly available and can be found at this repository.

**Benchmark on synthetic data** To match the asynchronous pulling commonly used in the literature, we consider that only one agent, randomly chosen by the environment, pulls an arm at each iteration. Similarly, to match the common assumption of the clustering structure, we consider Assumption 6.1 for the experiments. We propose to benchmark the algorithm on two variations of a synthetic environment. We consider $M = 3$ clusters of the

| Cumul. regret | $\Delta_{\theta,2}^{\min} = 0.2$ | $\Delta_{\theta,2}^{\min} = 1.27$ |
|---|---|---|
| Ind | 6100.7 +/- 468.4 | 5465.0 +/- 352.7 |
| *Oracle* | 318.5 +/- 684.0 | 292.9 +/- 914.3 |
| DynUCB | 5765.5 +/- 10439.5 | 15081.6 +/- 15299.7 |
| CLUB ($\gamma = \Delta_{\theta,2}^{\min}/4$) | 5772.5 +/- 532.4 | 5822.7 +/- 430.6 |
| CLUB ($\gamma = \Delta_{\theta,2}^{\min}/2$) | 5739.4 +/- 521.5 | 6138.1 +/- 659.8 |
| SCLUB ($\gamma = \Delta_{\theta,2}^{\min}/4$) | 6038.5 +/- 744.8 | 5949.3 +/- 501.0 |
| SCLUB ($\gamma = \Delta_{\theta,2}^{\min}/2$) | 6165.6 +/- 366.0 | 5775.3 +/- 416.8 |
| BASS ($\delta = 0.1$) | **1096.8 +/- 411.9** | 1617.5 +/- 689.8 |
| BASS ($\delta = 0.9$) | 1140.5 +/- 641.4 | **1563.2 +/- 750.7** |
| CMLB ($\gamma = \Delta_{\theta,2}^{\min}/4$) | 5895.3 +/- 382.5 | 5372.1 +/- 7575.0 |
| CMLB ($\gamma = \Delta_{\theta,2}^{\min}/2$) | 5918.1 +/- 431.3 | 20948.5 +/- 5355.5 |

*Table 2.* Comparison of the averaged cumulative regret last value $R_T$ for the different synthetic environments.

same size with bandit parameter $(\boldsymbol{\theta}_m)_{m \in \{1,...,M\}}$ defined as: $\forall 1 \leq q \leq \lceil \frac{M}{2} \rceil \boldsymbol{\theta}_{2q-1} = \boldsymbol{e}_q$, with $(\boldsymbol{e}_i)_i$ being the canonical basis and $\boldsymbol{\theta}_{2q}$ having its $q$-th entry being $\cos\omega$, its $(q+1)$-th entry being $\sin\omega$ and all the other entries being 0, with $\omega \in \{\pi/16, 7\pi/16\}$. The angle $\omega$ induces a problem complexity $\Delta_2^{\min}$, denoted $D$ in Figure 4, of 0.2 in the first scenario and 1.27 in the second.

We consider $K = 5 \times M$ arms of dimension $d = 10$. The algorithms are set to iterate for $T = 50000$. We set the number of agents to $N = 100$ and corrupt the reward observation with a centered and normalized Gaussian noise. As baselines, we choose *Ind*, which proposes to run the agents independently and *Oracle* which knows the true group structure and shares observations within the same cluster. We also compare our performance to state-of-the-art algorithms: *DynUCB*, *CLUB*, *SCLUB*, *CMLB*. For each algorithm, we consider two sets of hyper-parameters for which the result is shown in the 'dashed' and 'solid' lines. For each method, we line-search the $\alpha$ parameter. We set our hyperparameter $\gamma$ to 2 to match Gilitschenski & Hanebeck (2012). The experiment is run 50 times to average across runs. We detail the experimental setting and a description of the concurrent algorithms in Appendix R.

We report an additional case with $M = 6$ and an additional synthetic benchmark with Gaussian arms and bandit parameters in Appendix S.

In Table 2 (resp. Figure 3), we report the cumulative regret last value $R_T$ (resp. $R_t$) for all concurrent and baseline methods. First, we note the expected behavior of the baselines: *Oracle* provides the best performance for all scenarios, and the *Ind* algorithm provides intermediate performance, as the agent does not share its observations, but avoids introducing a detrimental estimation bias. We find that our approach, *BASS*, significantly outperforms all the other algorithms. In particular, we notice that our approach always results in a gain in comparison to *Ind*, as underlined by Section 6. Moreover, in Figure 4 we show the cumulative regret last value, $R_T$, of the evolution w.r.t. the UCB parameter $\alpha$. We

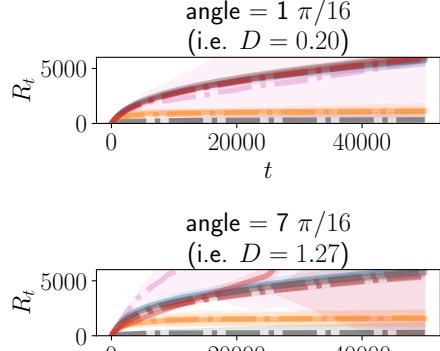

*Figure 3.* Comparison of the averaged evolution of the cumulative regret last value $R_t$ for the different synthetic environments considered.

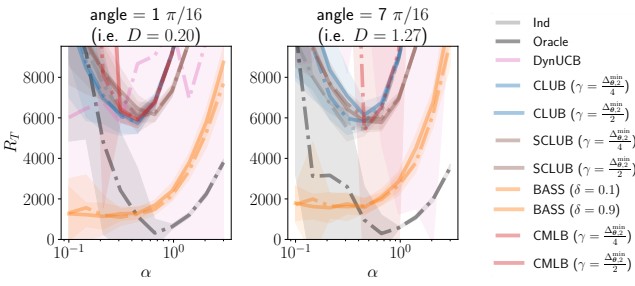

*Figure 4.* Comparison of the averaged evolution of the cumulative regret last value $R_T$ w.r.t. the UCB parameter $\alpha$ for the different synthetic environments considered.

can see that our approach systematically provides a lower regret than the state-of-the-art algorithms, underscoring the robustness of *BASS*.

In Table 3, we show the final clustering score value for all clustering algorithms. To quantify the degree of agreement between the estimated clusters/neighborhoods and the ground truth, we compute the $F_1$-score on the graph $\mathcal{G}_t$ w.r.t. the true cluster graph $\mathcal{G}$. Details of the score computation are reported in Appendix R. We observe that our approach always achieves the best performances. Moreover, most of the concurrent methods recover the clustering structure poorly. Since Nguyen & Lauw (2014); Gentile et al. (2014); Li et al. (2019); Ghosh et al. (2022) do not provide any experiment to verify this aspect, these results are not surprising.

**Benchmark on real data** To complete the performance study of our method, we propose a second benchmark with real datasets. We consider two public datasets of ranking scenarios: *MovieLens* and *Yahoo!* dataset. Since *Oracle* is not available for real data, we consider all other previous algorithms except it, keeping the same experimental setting. Details of the experimental setting are reported in Appendix R.

| Clustering score | $\Delta_{\theta,2}^{\min} = 0.2$ | $\Delta_{\theta,2}^{\min} = 1.27$ |
|---|---|---|
| Ind | 0.0 +/- 0.0 | 0.0 +/- 0.0 |
| *Oracle* | 1.0 +/- 0.0 | 1.0 +/- 0.0 |
| DynUCB | **0.7 +/- 0.1** | **0.9 +/- 0.2** |
| CLUB ($\gamma = \Delta_{\theta,2}^{\min}/4$) | 0.0 +/- 0.0 | 0.0 +/- 0.0 |
| CLUB ($\gamma = \Delta_{\theta,2}^{\min}/2$) | 0.0 +/- 0.0 | 0.0 +/- 0.0 |
| SCLUB ($\gamma = \Delta_{\theta,2}^{\min}/4$) | 0.0 +/- 0.0 | 0.0 +/- 0.0 |
| SCLUB ($\gamma = \Delta_{\theta,2}^{\min}/2$) | 0.0 +/- 0.0 | 0.0 +/- 0.0 |
| BASS ($\delta = 0.1$) | **0.7 +/- 0.0** | **0.9 +/- 0.0** |
| BASS ($\delta = 0.9$) | **0.7 +/- 0.0** | 0.8 +/- 0.0 |
| CMLB ($\gamma = \Delta_{\theta,2}^{\min}/4$) | 0.0 +/- 0.0 | 0.0 +/- 0.2 |
| CMLB ($\gamma = \Delta_{\theta,2}^{\min}/2$) | 0.0 +/- 0.0 | 0.6 +/- 0.1 |

*Table 3.* Comparison of the averaged F1 score of the estimated graph $\mathcal{G}_T$ for the different synthetic environments.

| Cumul. regret | Movie Lens | Yahoo |
|---|---|---|
| Ind | 11825.2 +/- 594.1 | 14800.5 +/- 876.0 |
| DynUCB ($M = 5$) | 16835.0 +/- 5429.9 | 14689.5 +/- 1912.7 |
| DynUCB ($M = 20$) | 17256.0 +/- 2314.2 | 16860.5 +/- 1625.3 |
| CLUB ($\gamma = 0.1$) | 11504.2 +/- 509.3 | 13321.5 +/- 4228.8 |
| CLUB ($\gamma = 1.0$) | 11804.8 +/- 562.4 | 13536.5 +/- 6548.2 |
| SCLUB ($\gamma = 0.1$) | 11425.2 +/- 764.0 | 13663.0 +/- 1444.0 |
| SCLUB ($\gamma = 1.0$) | 11085.5 +/- 701.3 | 13751.5 +/- 820.1 |
| BASS ($\delta = 0.1$) | **10273.2 +/- 686.3** | **12456.0 +/- 2096.1** |
| BASS ($\delta = 0.9$) | 10840.2 +/- 831.7 | 12533.5 +/- 1959.5 |
| CMLB ($\gamma = 0.1$) | 14944.2 +/- 341.6 | 16812.5 +/- 405.1 |
| CMLB ($\gamma = 1.0$) | 12010.8 +/- 669.4 | 16757.5 +/- 607.2 |

*Table 4.* Comparison of the averaged cumulative regret last value $R_T$ for the two datasets.

In Table 4, we show the final cumulative regret $R_T$ for the considered datasets. We observe that our approach again outperforms all the others in both scenarios. The overall performance comparison between the algorithms remains consistent with the synthetic experiment and confirms the good behavior of our algorithm; additional results are reported in Appendix S.

## 8. Conclusion

This paper explores collaboration as a sample sharing process, highlighting the benefits of introducing a bias, provided that it remains smaller than the current uncertainty. We derive a collaboration condition and propose an algorithm that takes advantage of sample sharing. Our approach has been extensively analyzed both theoretically and empirically, highlighting the key quantities involved in the process and demonstrating strong empirical performance on both synthetic and real-world datasets.

To our knowledge, this is the first work to have comprehensively analyzed the introduction of bias in this context or established an efficient condition and collaboration scheme with anisotropic testing. In line with these research directions, designing an arm-pulling strategy aimed at efficiently estimating the similarities could be a promising avenue.

## Acknowledgements

The authors would like to thank Pierre Ablin and Ievgen Redko for their insightful comments and valuable suggestions, which helped improve the quality and clarity of this work. We also thank anonymous reviewers for their constructive feedback during the review process.

## Impact Statement

This paper presents work whose goal is to advance the field of Machine Learning. There are many potential societal consequences of our work, none of which we feel must be specifically highlighted here.

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

# A. Appendix: table of contents

# B. Appendix: Notations summary

| Variables and parameters | Details |
| --- | --- |
| $d \in \mathbf{N}$ | Dimension |
| $K \in \mathbf{N}$ | Number of arms |
| $N \in \mathbf{N}$ | Number of agents |
| $M \in \mathbf{N}$ | True number of clusters |
| $\rho_i N \in \mathbf{N}$ | Number of agents within the neighborhood of agent $i$ |
| $T \in \mathbf{N}$ | Total number of iterations |
| $T_s(i,j) \in \mathbf{N}$ | Separation iteration of agent $i$ and $j$ |
| $T_c = \max_j T_s(i,j)$ | Iteration at which the cluster is adequately estimated |
| $\mathcal{X} = \{\forall k \in [K] \mid \boldsymbol{x}_k \in \mathbf{R}^d \text{ with } \|\boldsymbol{x}\|_2 \leq 1\}$ | Arm set |
| $\boldsymbol{\theta}_i^* \in \mathbf{R}^d$ | True bandit parameter of agent $i$ |
| $L \in \mathbf{R}^{*+}$ | Bound of the bandit parameter norm *i.e.*, $\forall i \; \|\boldsymbol{\theta}_i^*\|_2 \leq L$ |
| $\eta \in \mathbf{R}$ | Reward observation noise |
| $R \in \mathbf{R}^{*+}$ | Sub-Gaussianity parameter of the noise |
| $y_t = \boldsymbol{x}_t^\top \boldsymbol{\theta}^* + \eta \in \mathbf{R}$ | Reward signal at iteration t |
| $\mathcal{F}_t$ | $\sigma$-algebra generated by $(\boldsymbol{x}_1, y_1, \ldots, \boldsymbol{x}_{t-1}, y_{t-1}, \boldsymbol{x}_t)$ |
| $r_t = \max_k \boldsymbol{\theta}^{*\top} \boldsymbol{x}_k - \boldsymbol{\theta}^{*\top} \boldsymbol{x}_t \in \mathbf{R}^+$ | Instantaneous regret |
| $\boldsymbol{A}_{i,t} = \sum_{s=1}^t \boldsymbol{x}_{i,s} \boldsymbol{x}_{i,s}^\top \in \mathbf{R}^{d \times d}$ | Design matrix at iteration $t$ of agent $i$ |
| $\boldsymbol{A}_{\widehat{\mathcal{N}}_{i_t,t-1}} = \boldsymbol{A}_{i,0} + \sum_{j \in \hat{\mathcal{N}}_{i,t-1}} (\boldsymbol{A}_{j,t-1} - \boldsymbol{A}_{j,0}) \in \mathbf{R}^{d \times d}$ | Design matrix at iteration $t$ of neighborhood $i$ |
| $\boldsymbol{b}_{i,t} = \sum_{s=1}^t +y_{i,t} \boldsymbol{x}_{i,t} \in \mathbf{R}^d$ | Regressand for the estimation of the bandit parameter of the agent $i$ at iteration $t$ |
| $\boldsymbol{b}_{\widehat{\mathcal{N}}_{i_t,t-1}} = \sum_{j \in \widehat{\mathcal{N}}_{i_t,t-1}} \boldsymbol{b}_{j,t-1} \in \mathbf{R}^d$ | Regressand for the estimation of the bandit parameter of the neighborhood $i$ at iteration $t$ |
| $\widehat{\boldsymbol{\theta}}_{\widehat{\mathcal{N}}_{i_t,t-1}} = \boldsymbol{A}_{\widehat{\mathcal{N}}_{i_t,t-1}}^{-1} \boldsymbol{b}_{\widehat{\mathcal{N}}_{i_t,t-1}} \in \mathbf{R}^d$ | Bandit parameter estimation from neighborhood $i$ |
| $\hat{\boldsymbol{\theta}}_i \in \mathbf{R}^d$ | Bandit parameter estimation from agent $i$ |
| $\delta$ | Confidence level parameter |
| $\mathcal{C}_\delta(\hat{\boldsymbol{\theta}}_t)$ | Confidence region of the estimated bandit parameter |
| $\beta(\delta, t) = R\sqrt{2 \log \frac{1}{\delta} + \log \det(\boldsymbol{A}_t)} \in \mathbf{R}^{*+}$ | Confidence level of set of the previous region |
| $\widehat{\mathcal{N}}_i(t) = \{j \mid \Psi(i,j) = 1\}$ | Neighborhood of agent $i$ |
| $N_i^e(t)$ | Number of agents wrongly assigned to neighborhood $i$ at iteration $t$ |
| $\Delta_2^{i,j} = \|\boldsymbol{\theta}_i^* - \boldsymbol{\theta}_j^*\|_2 \in \mathbf{R}^{*+}$ | Bandit parameters Euclidean distance |
| $\Delta_2^{\min} = \min_{i,j \; i \neq j} \|\boldsymbol{\theta}_i^* - \boldsymbol{\theta}_j^*\|_2 \in \mathbf{R}^{*+}$ | Isotrope complexity of the problem |
| $\Delta_M^{i,j} = \|\boldsymbol{\theta}_i^* - \boldsymbol{\theta}_j^*\|_M \in \mathbf{R}^{*+}$ | Bandit parameters Mahalanobis distance weighted by matrix $\boldsymbol{M} \in \mathbf{R}^{d \times d}$ |
| $\Delta_M^{\min} = \min_{i,j \; i \neq j} \|\boldsymbol{\theta}_i^* - \boldsymbol{\theta}_j^*\|_M \in \mathbf{R}^{*+}$ | Anisotrope complexity of the problem weighted by matrix $\boldsymbol{M} \in \mathbf{R}^{d \times d}$ |
| $E_t = \{(i,j) \mid \Psi(i,j) = 0\}$ | Edges modelling the interaction between the agents |
| $V = \{1 \ldots N\}$ | Set of all agents |
| $G_t = (V, E_t)$ | Estimated agent graph at iteration $t$ |
| $\mathcal{I}_t \subset V$ | Agent pulling set at iteration $t$ |
| case $\mathcal{I}_t = \{i\}$ | Asynchronous pulling (with $i$ draw randomly from $V$) |
| case $\mathcal{I}_t = V$ | Synchronous pulling |

# C. Appendix: A simple two agents problem

## C.1. Preliminary theoretical results

We detail here the bounding of estimation error of $\boldsymbol{\theta}_1^*$. From the series of noisy projections $(y_{1,s})_{s=1}^t$, with $y_{1,s} = \boldsymbol{\theta}_1^{*\top} \boldsymbol{x}_{1,s} + \eta_{1,s}$ with $(\boldsymbol{x}_{1,s})_{s=1}^t \in \mathcal{X}^t$ a fixed series of arms. We consider the observation matrix to be invertible; the proof remains similar to the case involving the Moore-Penrose inverse. We can cancel the gradient of the OLS estimator to obtain:

$$\hat{\boldsymbol{\theta}}_t = \boldsymbol{\theta}_1^* + \boldsymbol{A}_t^{-1} \sum_{s=1}^t \boldsymbol{x}_{1,s} \eta_{1,s} \quad \text{with} \quad \boldsymbol{A}_t = \sum_{s=1}^t \boldsymbol{x}_{1,s} \boldsymbol{x}_{1,s}\top$$

$$\left\| \hat{\boldsymbol{\theta}}_t - \boldsymbol{\theta}_1^* \right\|_{\boldsymbol{A}_t} = \left\| \sum_{s=1}^t \boldsymbol{x}_{1,s} \eta_{1,s} \right\|_{\boldsymbol{A}_t^{-1}}$$

$$\left\| \hat{\boldsymbol{\theta}}_t - \boldsymbol{\theta}_1^* \right\|_{\boldsymbol{A}_t} \leq \beta(\delta, \boldsymbol{A}_t) \quad \text{as defined in Theorem 4.1}$$

Similarly, for the collaboration case we have:

$$\hat{\boldsymbol{\theta}}_t^{\text{collab}} = \frac{1}{2}(\boldsymbol{\theta}_1^* + \boldsymbol{\theta}_2^*) + \boldsymbol{A}_t^{-1} \sum_{s=1}^t \boldsymbol{x}_{1,s}(\eta_{1,s} + \eta_{2,s})$$

$$\left\| \hat{\boldsymbol{\theta}}_t^{\text{collab}} - \boldsymbol{\theta}_1^* \right\|_{\boldsymbol{A}_t} = \left\| \frac{1}{2}(\boldsymbol{\theta}_2^* - \boldsymbol{\theta}_1^*) + \frac{1}{2}\boldsymbol{A}_t^{-1} \sum_{s=1}^t \boldsymbol{x}_{1,s}(\eta_{1,s} + \eta_{2,s}) \right\|_{\boldsymbol{A}_t}$$

$$\left\| \hat{\boldsymbol{\theta}}_t^{\text{collab}} - \boldsymbol{\theta}_1^* \right\|_{\boldsymbol{A}_t} \leq \frac{1}{2} \left\| \boldsymbol{\theta}_2^* - \boldsymbol{\theta}_1^* \right\|_{\boldsymbol{A}_t} + \frac{1}{\sqrt{2}} \beta(\delta, \boldsymbol{A}_t) \quad \text{as defined in Theorem 4.1}$$

Using the two previous error bounds, we compute the ratio of the bounding terms and compare it to 1 to gain insight into the collaboration condition, leading to:

$$\frac{\frac{1}{2} \left\| \boldsymbol{\theta}_j^* - \boldsymbol{\theta}_i^* \right\|_{\boldsymbol{A}_t} + \frac{1}{\sqrt{2}} \beta(\delta, \boldsymbol{A}_t)}{\beta(\delta, \boldsymbol{A}_t)} \leq 1$$

$$\left\| \boldsymbol{\theta}_1^* - \boldsymbol{\theta}_2^* \right\|_{\boldsymbol{A}_t} \leq (2 - \sqrt{2}) \beta(\delta, \boldsymbol{A}_t)$$

## C.2. Empirical illustration

To generate Figure 1, we consider a simple case with $d = 2$ and a set of two arms $\mathcal{X} = \{\boldsymbol{e}_1, \boldsymbol{e}_2\}$, where $\boldsymbol{e}_1$ and $\boldsymbol{e}_2$ are the canonical basis vectors. We set $\boldsymbol{\theta}_1^* = \boldsymbol{e}_1$ and define $\boldsymbol{\theta}_2^* = \boldsymbol{\theta}_1^* + \boldsymbol{p}$, where $\boldsymbol{p}$ represents a significant perturbation. The noise is modeled as normalized Gaussian noise. We set $t = 10$, allowing us to gather 10 noisy projections for each bandit estimation problem, denoted by $(y_{1,s})_{s=1}^{10}$ and $(y_{2,s})_{s=1}^{10}$, with $y_{1,s} = \boldsymbol{\theta}_1^{*\top} \boldsymbol{x}_s + \eta_{1,s}$ (resp. $y_{2,s} = \boldsymbol{\theta}_2^{*\top} \boldsymbol{x}_s + \eta_{2,s}$). The design matrix is shared between both problems, i.e., $\boldsymbol{A}_t = \boldsymbol{A}_{1,t} = \boldsymbol{A}_{2,t} = \sum_{s=1}^{10} \boldsymbol{x}_s \boldsymbol{x}_s^\top$. At each iteration, the arm $\boldsymbol{x}_s$ is chosen at random uniformly.

## C.3. The Bandit Adaptive Sample Sharing algorithm

We include a simplified flow chart to clarify and summarize the main steps of our algorithm.

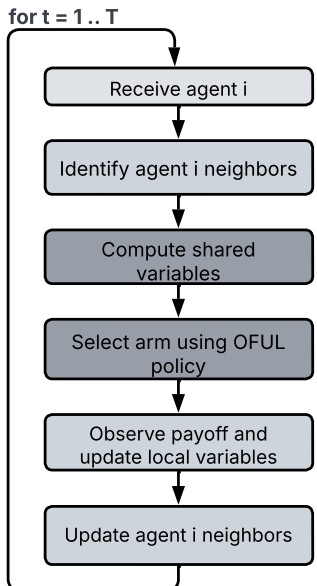

*Figure 5.* Overview of the Bandit Adaptive Sample Sharing (*BASS*) Algorithm

In Figure 5, we summarize the main steps of the BASS algorithm, illustrating the process from selecting an arm using the OFUL policy, to identifying the agent's neighborhood, updating shared and local variables, and updating the agent graph.

## D. Appendix: Technical lemmas

We gather here technical lemmas used later in the remaining proofs.

**Lemma D.1** (Error vector of the difference of two estimated bandit parameters)**.** *At iteration $t$, considering the agent $i, j$ and their corresponding bandit parameter estimation $\widehat{\boldsymbol{\theta}}_{i,t}$, $\widehat{\boldsymbol{\theta}}_{i,t}$, with $\boldsymbol{\theta}_i^* \neq \boldsymbol{\theta}_j^*$, we have:*

$$\widehat{\boldsymbol{\theta}}_{i,t} - \widehat{\boldsymbol{\theta}}_{i,t} = \boldsymbol{\theta}_i^* - \boldsymbol{\theta}_j^* + \boldsymbol{A}_t^{-1} Z_{i,j,t} \quad \text{with } Z_{i,j} = \sum_{s=1}^{t} \boldsymbol{x}_{i,s} \eta_{i,s} - \boldsymbol{x}_{j,s} \eta_{j,s}$$

*Proof of* Lemma D.1. At iteration $t$, considering the agent $i$ and their corresponding bandit parameter estimation $\widehat{\boldsymbol{\theta}}_{i,t}$ with $\boldsymbol{\theta}_i^*$ its true bandit parameter, we have:

$$\widehat{\boldsymbol{\theta}}_{i,t} = \boldsymbol{\theta}_i^* + \boldsymbol{A}_{i,t}^{-1} \sum_{s=1}^{t} \boldsymbol{x}_{i,s} \eta_{i,s} \quad \text{with} \quad \boldsymbol{A}_{i,t} = \sum_{s=1}^{t} \boldsymbol{x}_{i,s} \boldsymbol{x}_{i,s}^\top \ ,$$

The last equation being obtained by cancelling the gradient associated with the OLS. Similarly, if we consider a second agent and their difference, we have:

$$\widehat{\boldsymbol{\theta}}_{i,t} - \widehat{\boldsymbol{\theta}}_{j,t} = \boldsymbol{\theta}_i^* - \boldsymbol{\theta}_j^* + \boldsymbol{A}_{i,t}^{-1} \sum_{s=1}^{t} \boldsymbol{x}_{i,s} \eta_{i,s} + \boldsymbol{A}_{j,t}^{-1} \sum_{s=1}^{t} \boldsymbol{x}_{j,s} \eta_{j,s}$$

$$= \boldsymbol{\theta}_i^* - \boldsymbol{\theta}_j^* + \boldsymbol{A}_t^{-1} \sum_{s=1}^{t} \boldsymbol{x}_{i,s} \eta_{i,s} - \boldsymbol{x}_{j,s} \eta_{j,s} \quad \text{since synchronous pulling, we have} \quad \boldsymbol{A}_{i,t} = \boldsymbol{A}_{j,t} = \boldsymbol{A}_t$$

$$= \boldsymbol{\theta}_i^* - \boldsymbol{\theta}_j^* + \boldsymbol{A}_t^{-1} Z_{i,j,t} \quad \text{with } Z_{i,j} = \sum_{s=1}^{t} \boldsymbol{x}_{i,s} \eta_{i,s} - \boldsymbol{x}_{j,s} \eta_{j,s}$$

$\square$

**Definition D.1** (Optimistic true bandit parameter). *Let $\beta > 0$, $\widehat{\boldsymbol{\theta}} \in \mathbf{R}^d$, $\boldsymbol{x} \in \mathbf{R}^d$ and $\boldsymbol{A} \in \mathbf{R}^{d \times d}$ a positive semi-definite matrix, we define by the* optimistic true bandit parameter, *the vector $\tilde{\boldsymbol{\theta}}$ defined as:*

$$\tilde{\boldsymbol{\theta}} = \arg\max_{\boldsymbol{\theta} \in \mathbf{R}^d} \boldsymbol{\theta}^\top \boldsymbol{x} \quad such \; as \quad \|\boldsymbol{\theta} - \widehat{\boldsymbol{\theta}}\|_{\boldsymbol{A}}^2 \leq \beta^2$$

**Lemma D.2** (Characterization of the optimistic true bandit parameter). *Let $\beta > 0$, $\widehat{\boldsymbol{\theta}} \in \mathbf{R}^d$, $\boldsymbol{x} \in \mathbf{R}^d$, $\boldsymbol{A} \in \mathbf{R}^{d \times d}$ a positive semi-definite matrix and $\tilde{\boldsymbol{\theta}} \in \mathbf{R}^d$ the corresponding optimistic true bandit parameter is:*

$$\tilde{\boldsymbol{\theta}} = \widehat{\boldsymbol{\theta}} + \frac{\beta}{\|\boldsymbol{x}\|_{\boldsymbol{A}^{-1}}} \boldsymbol{A}^{-1} \boldsymbol{x}$$

*Proof of* Lemma D.2. Let consider $\beta > 0$, $\widehat{\boldsymbol{\theta}} \in \mathbf{R}^d$, $\boldsymbol{x} \in \mathbf{R}^d$, $\boldsymbol{A} \in \mathbf{R}^{d \times d}$ a positive semi-definite matrix, the corresponding $\tilde{\boldsymbol{\theta}} \in \mathbf{R}^d$ and the following optimization problem:

$$\tilde{\boldsymbol{\theta}} = \arg\max_{\boldsymbol{\theta} \in \mathbf{R}^d} \boldsymbol{\theta}^\top \boldsymbol{x} \quad such \; as \quad \|\boldsymbol{\theta} - \widehat{\boldsymbol{\theta}}\|_{\boldsymbol{A}}^2 \leq \beta^2$$

We derive the corresponding Lagrandian: $\mathcal{L}(\boldsymbol{\theta}, \mu) = \boldsymbol{\theta}^\top \boldsymbol{x} - \mu(\|\boldsymbol{\theta} - \widehat{\boldsymbol{\theta}}\|_{\boldsymbol{A}}^2 - \beta^2)$ and cancelling the associated gradient is equivalent to:

$$\iff \begin{cases} \boldsymbol{A}\tilde{\boldsymbol{\theta}} = \frac{1}{2\mu}\boldsymbol{x} + \boldsymbol{A}\widehat{\boldsymbol{\theta}} \\ \beta^2 = \|\boldsymbol{\theta} - \widehat{\boldsymbol{\theta}}\|_{\boldsymbol{A}}^2 \end{cases}$$

$$\iff \begin{cases} \tilde{\boldsymbol{\theta}} = \frac{1}{2\mu}\boldsymbol{A}^{-1}\boldsymbol{x} + \widehat{\boldsymbol{\theta}} \\ \|\boldsymbol{x}\|_{\boldsymbol{A}^{-1}}^2 = 4\mu^2\beta^2 \end{cases}$$

$$\iff \tilde{\boldsymbol{\theta}} = \widehat{\boldsymbol{\theta}} + \frac{\beta}{\|\boldsymbol{x}\|_{\boldsymbol{A}^{-1}}} \boldsymbol{A}^{-1}\boldsymbol{x}$$

$\square$

**Lemma D.3** (Positivity of the clustering error). *For each agent $i$, at probability $1 - \delta$, we have:*

$$\forall t > 0 \quad we \; have: \quad \hat{N}_{i,t} - \rho_{m(i)} N \geq 0 \; .$$

*Proof of* Lemma D.3. By using proof by contradiction, at iteration $t > 0$, let us consider an agent $i$ and suppose that $\hat{N}_{i,t} \leq \rho_{m(i)} N$. This implies there exists an agent $j$, such as $\Psi(i, j, t) = 1$ and $\boldsymbol{\theta}_j^* = \boldsymbol{\theta}_i^*$. Let us develop $\Psi(i, j, t) = 1$, at probability $1 - \delta$, we have:

$$\|\hat{\boldsymbol{\theta}}_i - \hat{\boldsymbol{\theta}}_j\|_{\boldsymbol{A}_t} \geq 2\beta(\delta, \boldsymbol{A}_t)$$
$$\|\boldsymbol{\theta}_i^* - \boldsymbol{\theta}_j^*\|_{\boldsymbol{A}_t} + \|\boldsymbol{Z}_{i,j,t}\|_{\boldsymbol{A}_t^{-1}} \geq 2\beta(\delta, \boldsymbol{A}_t)$$
$$\sqrt{2}\beta(\delta, \boldsymbol{A}_t) \geq 2\beta(\delta, \boldsymbol{A}_t)$$

Hence, for each agent $i$, at iteration $t > 0$, with probability $1 - \delta$: $\hat{N}_{i,t} \geq \rho_{m(i)} N$

$\square$

**Lemma D.4** ($\beta$ ratio upper-bound). *Let $\delta > 0$, $N \in \mathbf{N}$, $t > 0$ and $\boldsymbol{A}_t$ the design matrix obtained after $t$ iterations for a given agent, we have:*

$$\frac{\beta(\delta, N\boldsymbol{A}_t)}{\beta(\delta, \boldsymbol{A}_t)} \leq \sqrt{1 + \frac{d \log N}{2 \log \frac{1}{\delta}}}$$

*Proof of* Lemma D.4. Let us consider $\delta > 0$, $N \in \mathbf{N}$, $t > 0$ and $\boldsymbol{A}_t$ the design matrix of a given agent, we have:

$$
\begin{aligned}
\frac{\beta(\delta, N\boldsymbol{A}_t)}{\beta(\delta, \boldsymbol{A}_t)} &= \sqrt{\frac{2\log\frac{1}{\delta} + \log\det(N\boldsymbol{A}_t)}{2\log\frac{1}{\delta} + \log\det(\boldsymbol{A}_t)}} \\
&= \sqrt{\frac{2\log\frac{1}{\delta} + d\log N + \log\det(\boldsymbol{A}_t)}{2\log\frac{1}{\delta} + \log\det(\boldsymbol{A}_t)}} \\
&\leq \sqrt{1 + \frac{d\log N}{2\log\frac{1}{\delta}}} \quad \text{since} \quad \det(\boldsymbol{A}_t) \geq \det(\boldsymbol{A}_{t-1}) \quad \text{and} \quad \boldsymbol{A}_0 = \boldsymbol{I}
\end{aligned}
$$

$\square$

# E. Appendix: Bandit parameter estimation error bounding

**Theorem 4.1** (Confidence Ellipsoid for Bandit Parameter Estimation). *Let $\delta \in (0, 1)$, $t > 0$, and $\boldsymbol{\theta}^* \in \mathbb{R}^d$. Let $(\boldsymbol{x}_s)_{1 \leq s \leq t}$ denote the sequence of arms pulled up to time $t$. Under* Assumption 4.1 *and* Assumption 4.2, *the following holds with probability at least $1 - \delta$:*

$$
\mathbb{P}\left(\boldsymbol{\theta}^* \in \mathcal{C}_\delta(\hat{\boldsymbol{\theta}}_t)\right) \geq 1 - \delta \ ,
$$

*where*

$$
\begin{cases}
\mathcal{C}_\delta(\hat{\boldsymbol{\theta}}_t) = \left\{ \left\| \hat{\boldsymbol{\theta}}_t - \boldsymbol{\theta}^* \right\|_{\boldsymbol{A}_t} \leq \beta(\delta, \boldsymbol{A}_t) \right\} \ , \\
\beta(\delta, \boldsymbol{A}_t) = R\sqrt{2\log\left(\frac{1}{\delta}\sqrt{\frac{\det(\boldsymbol{A}_t)}{\det(\boldsymbol{A}_0)}}\right)} \ ,
\end{cases}
\tag{2}
$$

*and $\boldsymbol{A}_t = \sum_{s=1}^{t} \boldsymbol{x}_s \boldsymbol{x}_s^\top$ is the empirical design matrix.*

*Proof of* Theorem 4.1. The proof can be founded in the supplementary material of Abbasi-Yadkori et al. (2011).

$\square$

# F. Appendix: Instantaneous regret upper bounds

**Lemma 5.1** (Instantaneous Regret Upper Bounds). *Let $0 < \delta < 1$ and $t > 0$. Let $(\boldsymbol{\theta}_j^*)_{1 \leq j \leq N} \in \mathbb{R}^{d \times N}$ be the true parameters of $N$ linear bandit models. Denote by $\boldsymbol{x}_{i,t}$ (resp. $\boldsymbol{x}_{\widehat{\mathcal{N}}_i(t)}$) the arm selected by agent $i$ using the local (resp. collaborative) strategy. Then, with probability at least $1 - \delta$, the instantaneous regret of agent $i$ is bounded as follows:*

*(i) Local strategy:*

$$
r_i(t) \leq 2\beta\left(\delta, \boldsymbol{A}_{i,t}\right) \|\boldsymbol{x}_{i,t}\|_{\boldsymbol{A}_{i,t}^{-1}} \ .
$$

*(ii) Collaborative strategy:*

$$
r_i(t) \leq \frac{2\beta(\delta, m\boldsymbol{A}_{i,t})}{\sqrt{m}} \|\boldsymbol{x}_{\widehat{\mathcal{N}}_i(t)}\|_{\boldsymbol{A}_{i,t}^{-1}} + \frac{2}{m} \sum_{j \in \widehat{\mathcal{N}}_i(t)} \Delta_{\boldsymbol{A}_{i,t}}^{i,j} \ ,
$$

*where $m = |\widehat{\mathcal{N}}_i(t)|$ and $\Delta_{\boldsymbol{A}_{i,t}}^{i,j} = \left\|\boldsymbol{\theta}_i^* - \boldsymbol{\theta}_j^*\right\|_{\boldsymbol{A}_{i,t}}$.*

*Proof of* Lemma 5.1. For the single case, at round $t$, for a given agent $i$, by considering the optimistic arm $\boldsymbol{x}_{i,t}$ and the optimal arm $\boldsymbol{x}_i^*$, we have:

$$
\begin{aligned}
r_{i,t} &= \boldsymbol{\theta}_i^{*\top}(\boldsymbol{x}_i^* - \boldsymbol{x}_{i,t}) \\
&= \boldsymbol{\theta}_i^{*\top}\boldsymbol{x}_i^* - \boldsymbol{\theta}_i^{*\top}\boldsymbol{x}_{i,t} \\
&\leq \tilde{\boldsymbol{\theta}}_{i,t}^{\top}\boldsymbol{x}_{i,t} - \boldsymbol{\theta}_i^{*\top}\boldsymbol{x}_{i,t} \quad \text{since } (\tilde{\boldsymbol{\theta}}_{i,t}, \boldsymbol{x}_{i,t}) \quad \text{are the optimistic tuple} \\
&\leq \left(\widehat{\boldsymbol{\theta}}_{i,t} + \frac{\beta(\delta, \boldsymbol{A}_t)}{\|\boldsymbol{x}_{i,t}\|_{\boldsymbol{A}_t^{-1}}}\boldsymbol{A}_t^{-1}\boldsymbol{x}_{i,t} - \boldsymbol{\theta}_i^*\right)^{\top}\boldsymbol{x}_{i,t} \quad \text{from Lemma D.2} \\
&\leq \left\|\widehat{\boldsymbol{\theta}}_{i,t} - \boldsymbol{\theta}_i^*\right\|_{\boldsymbol{A}_t}\|\boldsymbol{x}_{i,t}\|_{\boldsymbol{A}_t^{-1}} + \frac{\beta(\delta, \boldsymbol{A}_t)}{\|\boldsymbol{x}_{i,t}\|_{\boldsymbol{A}_t^{-1}}}\|\boldsymbol{x}_{i,t}\|_{\boldsymbol{A}_t^{-1}}^2 \\
&\leq 2\beta(\delta, \boldsymbol{A}_t)\|\boldsymbol{x}_{i,t}\|_{\boldsymbol{A}_t^{-1}}
\end{aligned}
$$

Moreover, for the collaborative case, at round $t$, for a given agent $i$, by considering the optimistic arm $\boldsymbol{x}_{\widehat{\mathcal{N}}_i(t)}$ and the optimal arm $\boldsymbol{x}_i^*$, we have:

$$
\begin{aligned}
r_{i,t} &= \boldsymbol{\theta}_i^{*\top}(\boldsymbol{x}_i^* - \boldsymbol{x}_{\widehat{\mathcal{N}}_i(t)}) \\
&= (\boldsymbol{\theta}_{\widehat{\mathcal{N}}_i(t)}^* - \boldsymbol{\theta}_{\widehat{\mathcal{N}}_i(t)}^* + \boldsymbol{\theta}_i^*)^{\top}(\boldsymbol{x}_i^* - \boldsymbol{x}_{\widehat{\mathcal{N}}_i(t)}) \\
&= \boldsymbol{\theta}_{\widehat{\mathcal{N}}_i(t)}^{*\top}\boldsymbol{x}_i^* - \boldsymbol{\theta}_{\widehat{\mathcal{N}}_i(t)}^{*\top}\boldsymbol{x}_{\widehat{\mathcal{N}}_i(t)} + (\boldsymbol{\theta}_i^* - \boldsymbol{\theta}_{\widehat{\mathcal{N}}_i(t)}^*)^{\top}(\boldsymbol{x}_i^* - \boldsymbol{x}_{\widehat{\mathcal{N}}_i(t)}) \\
&\leq \tilde{\boldsymbol{\theta}}_{\widehat{\mathcal{N}}_i(t)}^{\top}\boldsymbol{x}_{\widehat{\mathcal{N}}_i(t)} - \boldsymbol{\theta}_{\widehat{\mathcal{N}}_i(t)}^{*\top}\boldsymbol{x}_{\widehat{\mathcal{N}}_i(t)} + (\boldsymbol{\theta}_i^* - \boldsymbol{\theta}_{\widehat{\mathcal{N}}_i(t)}^*)^{\top}(\boldsymbol{x}_i^* - \boldsymbol{x}_{\widehat{\mathcal{N}}_i(t)}) \quad \text{since } (\tilde{\boldsymbol{\theta}}_{\widehat{\mathcal{N}}_i(t)}, \boldsymbol{x}_{\widehat{\mathcal{N}}_i(t)}) \quad \text{are the optimistic tuple} \\
&\leq (\tilde{\boldsymbol{\theta}}_{\widehat{\mathcal{N}}_i(t)} - \boldsymbol{\theta}_{\widehat{\mathcal{N}}_i(t)}^*)^{\top}\boldsymbol{x}_{\widehat{\mathcal{N}}_i(t)} + (\boldsymbol{\theta}_i^* - \boldsymbol{\theta}_{\widehat{\mathcal{N}}_i(t)}^*)^{\top}(\boldsymbol{x}_i^* - \boldsymbol{x}_{\widehat{\mathcal{N}}_i(t)})
\end{aligned}
$$
(5)

We now consider the corresponding norms, we have:

$$
\begin{aligned}
&\leq (\tilde{\boldsymbol{\theta}}_{\widehat{\mathcal{N}}_i(t)} - \boldsymbol{\theta}_{\widehat{\mathcal{N}}_i(t)}^*)^{\top}\boldsymbol{x}_{\widehat{\mathcal{N}}_i(t)} + \|\boldsymbol{\theta}_i^* - \boldsymbol{\theta}_{\widehat{\mathcal{N}}_i(t)}^*\|_{\boldsymbol{A}_t}\|\boldsymbol{x}_i^* - \boldsymbol{x}_{\widehat{\mathcal{N}}_i(t)}\|_{\boldsymbol{A}_t^{-1}} \\
&\leq (\tilde{\boldsymbol{\theta}}_{\widehat{\mathcal{N}}_i(t)} - \boldsymbol{\theta}_{\widehat{\mathcal{N}}_i(t)}^*)^{\top}\boldsymbol{x}_{\widehat{\mathcal{N}}_i(t)} + 2\|\boldsymbol{\theta}_{\widehat{\mathcal{N}}_i(t)}^* - \boldsymbol{\theta}_i^*\|_{\boldsymbol{A}_t} \quad \text{from Assumption 4.1} \\
&\leq \left(\widehat{\boldsymbol{\theta}}_{\widehat{\mathcal{N}}_i(t)} + \frac{\beta(\delta, \boldsymbol{A}_{\widehat{\mathcal{N}}_i(t)})}{\|\boldsymbol{x}_{\widehat{\mathcal{N}}_i(t)}\|_{\boldsymbol{A}_{\widehat{\mathcal{N}}_i(t)}^{-1}}}\boldsymbol{A}_{\widehat{\mathcal{N}}_i(t)}^{-1\top}\boldsymbol{x}_{\widehat{\mathcal{N}}_i(t)} - \boldsymbol{\theta}_{\widehat{\mathcal{N}}_i(t)}^*\right)^{\top}\boldsymbol{x}_{\widehat{\mathcal{N}}_i(t)} + 2\|\boldsymbol{\theta}_{\widehat{\mathcal{N}}_i(t)}^* - \boldsymbol{\theta}_i^*\|_{\boldsymbol{A}_t} \quad \text{from Lemma D.2} \\
&\leq \left(\frac{\beta(\delta, \boldsymbol{A}_{\widehat{\mathcal{N}}_i(t)})}{\|\boldsymbol{x}_{\widehat{\mathcal{N}}_i(t)}\|_{\boldsymbol{A}_{\widehat{\mathcal{N}}_i(t)}^{-1}}}\boldsymbol{A}_{\widehat{\mathcal{N}}_i(t)}^{-1\top}\boldsymbol{x}_{\widehat{\mathcal{N}}_i(t)}\right)^{\top}\boldsymbol{x}_{\widehat{\mathcal{N}}_i(t)} + \left(\widehat{\boldsymbol{\theta}}_{\widehat{\mathcal{N}}_i(t)} - \boldsymbol{\theta}_{\widehat{\mathcal{N}}_i(t)}^*\right)^{\top}\boldsymbol{x}_{\widehat{\mathcal{N}}_i(t)} + 2\|\boldsymbol{\theta}_{\widehat{\mathcal{N}}_i(t)}^* - \boldsymbol{\theta}_i^*\|_{\boldsymbol{A}_t} \\
&\leq \beta(\delta, \boldsymbol{A}_{\widehat{\mathcal{N}}_i(t)})\|\boldsymbol{x}_{\widehat{\mathcal{N}}_i(t)}\|_{\boldsymbol{A}_{\widehat{\mathcal{N}}_i(t)}^{-1}} + \|\widehat{\boldsymbol{\theta}}_{\widehat{\mathcal{N}}_i(t)} - \boldsymbol{\theta}_{\widehat{\mathcal{N}}_i(t)}^*\|_{\boldsymbol{A}_{\widehat{\mathcal{N}}_i(t)}}\|\boldsymbol{x}_{\widehat{\mathcal{N}}_i(t)}\|_{\boldsymbol{A}_{\widehat{\mathcal{N}}_i(t)}^{-1}} + 2\|\boldsymbol{\theta}_{\widehat{\mathcal{N}}_i(t)}^* - \boldsymbol{\theta}_i^*\|_{\boldsymbol{A}_t} \\
&\leq \left(\beta(\delta, \boldsymbol{A}_{\widehat{\mathcal{N}}_i(t)}) + \|\widehat{\boldsymbol{\theta}}_{\widehat{\mathcal{N}}_i(t)} - \boldsymbol{\theta}_{\widehat{\mathcal{N}}_i(t)}^*\|_{\boldsymbol{A}_{\widehat{\mathcal{N}}_i(t)}}\right)\|\boldsymbol{x}_{\widehat{\mathcal{N}}_i(t)}\|_{\boldsymbol{A}_{\widehat{\mathcal{N}}_i(t)}^{-1}} + 2\|\boldsymbol{\theta}_{\widehat{\mathcal{N}}_i(t)}^* - \boldsymbol{\theta}_i^*\|_{\boldsymbol{A}_t} \\
&\leq 2\beta(\delta, \boldsymbol{A}_{\widehat{\mathcal{N}}_i(t)})\|\boldsymbol{x}_{\widehat{\mathcal{N}}_i(t)}\|_{\boldsymbol{A}_{\widehat{\mathcal{N}}_i(t)}^{-1}} + 2\|\boldsymbol{\theta}_{\widehat{\mathcal{N}}_i(t)}^* - \boldsymbol{\theta}_i^*\|_{\boldsymbol{A}_t} \quad \text{by definition of } \beta(\delta, \boldsymbol{A}_{\widehat{\mathcal{N}}_i(t)}) \\
&\leq \frac{2}{\sqrt{N}}\beta(\delta, N\boldsymbol{A}_t)\|\boldsymbol{x}_{\widehat{\mathcal{N}}_i(t)}\|_{\boldsymbol{A}_t^{-1}} + \frac{2}{N}\sum_{j=1}^{N}\|\boldsymbol{\theta}_j^* - \boldsymbol{\theta}_i^*\|_{\boldsymbol{A}_t}
\end{aligned}
$$

$\square$

# G. Appendix: Collaboration condition

We proof Equation (3). With Assumption 4.1, we can bound the term $\|\boldsymbol{x}_{i,t}\|_{\boldsymbol{A}_t^{-1}}$ in Lemma 5.1, leading to:

$$
r_{i,t} \leq 2\beta(\delta, \boldsymbol{A}_t)
$$
(6)

Similarly, for the collaborating case of two agents, we have:

$$r_{i,t} \leq \sqrt{2}\beta(\delta, 2\boldsymbol{A}_t) + \|\boldsymbol{\theta}_j^* - \boldsymbol{\theta}_i^*\|_{\boldsymbol{A}_t} \tag{7}$$

Equation (6) and Equation (7) leads us to:

$$\frac{1}{\sqrt{2}}\frac{\beta(\delta, 2\boldsymbol{A}_t)}{\beta(\delta, \boldsymbol{A}_t)} + \frac{\|\boldsymbol{\theta}_j^* - \boldsymbol{\theta}_i^*\|_{\boldsymbol{A}_t}}{2\beta(\delta, \boldsymbol{A}_t)} \leq 1$$

$$\frac{\|\boldsymbol{\theta}_j^* - \boldsymbol{\theta}_i^*\|_{\boldsymbol{A}_t}}{2\beta(\delta, \boldsymbol{A}_t)} \leq 1 - \frac{1}{\sqrt{2}}\frac{\beta(\delta, 2\boldsymbol{A}_t)}{\beta(\delta, \boldsymbol{A}_t)}$$

$$\frac{\|\boldsymbol{\theta}_j^* - \boldsymbol{\theta}_i^*\|_{\boldsymbol{A}_t}}{2\beta(\delta, \boldsymbol{A}_t)} \leq 1 - \frac{1}{\sqrt{2}}$$

$$\|\boldsymbol{\theta}_j^* - \boldsymbol{\theta}_i^*\|_{\boldsymbol{A}_t} \leq (2 - \sqrt{2})\beta(\delta, \boldsymbol{A}_t)$$

## H. Appendix: Ellipsoid separation under synchronous pulling

**Lemma 5.2** (Ellipsoid Separation Under Synchronous Pulling). *Let $0 < \delta < 1$, and consider two agents $i$ and $j$. Under Assumption 4.1 and Assumption 4.2, define $\tilde{\beta} = (1 + 1/\sqrt{2})\beta(\delta, \boldsymbol{A}_t)$. Then the following statements are equivalent:*

*(i)* $\left\|\hat{\boldsymbol{\theta}}_i - \hat{\boldsymbol{\theta}}_j\right\|_{\boldsymbol{A}_t} \geq (2 + \sqrt{2})\beta(\delta, \boldsymbol{A}_t)$ ,

*(ii)* $\mathcal{E}(\hat{\boldsymbol{\theta}}_i, \boldsymbol{A}_t, \tilde{\beta}) \cap \mathcal{E}(\hat{\boldsymbol{\theta}}_j, \boldsymbol{A}_t, \tilde{\beta}) = \emptyset$ .

*Proof of* Lemma 5.2. The proof can be founded in the supplementary material of Gilitschenski & Hanebeck (2012).

$\square$

## I. Appendix: Ellipsoid separation test function

We prove the ellipsoid separation property of the $\Psi$ function.

**Property I.1** ($\kappa$ellipsoid separation property of the $\Psi$ function.). *Let $0 < \delta < 1$, for two agents $i$ and $j$, and following Assumption 4.1 and Assumption 4.2, the following statements are equivalent:*

$$\forall i, j, t > 0 \quad \Psi(i, j, t) = 1 \iff \mathcal{E}(\hat{\boldsymbol{\theta}}_i, \boldsymbol{A}_t, \tilde{\beta}) \cap \mathcal{E}(\hat{\boldsymbol{\theta}}_j, \boldsymbol{A}_t, \tilde{\beta}) = \emptyset ,$$

*Proof of* Property I.1. Let us consider two agents $i$ and $j$, with their associated unknown bandit parameter estimates $\hat{\boldsymbol{\theta}}_i$ and $\hat{\boldsymbol{\theta}}_j$, the observation matrices $\boldsymbol{A}_i$ and $\boldsymbol{A}_j$ along with their local confidence level briefly noted $\tilde{\beta}_i$ and $\tilde{\beta}_j$ as in (4.1). If we consider that $\boldsymbol{\Phi}$ and $\mathrm{diag}(\boldsymbol{\eta})$, the eigenvectors and the eigenvalues of the generalized eigenvalue problem, we have:

$$\boldsymbol{A}_i \boldsymbol{\Phi} = \boldsymbol{A}_j \boldsymbol{\Phi} \mathrm{diag}(\boldsymbol{\eta}) \quad \text{with} \quad \boldsymbol{\Phi}_i^\top \boldsymbol{A}_j \boldsymbol{\Phi}_i = 1$$

First, with the given scaling of this problem being, one can recover the following equivalent constraints:

$$\begin{cases} \boldsymbol{\Phi}^\top \boldsymbol{A}_i \boldsymbol{\Phi} = \mathrm{diag}(\boldsymbol{\eta}) \\ \boldsymbol{\Phi}^\top \boldsymbol{A}_j \boldsymbol{\Phi} = \boldsymbol{I} \end{cases}$$

Which gives us:

$$\begin{cases} \boldsymbol{A}_i^{-1} = \boldsymbol{\Phi}^\top \mathrm{diag}(\boldsymbol{\eta}) \boldsymbol{\Phi} \\ \boldsymbol{A}_j^{-1} = \boldsymbol{\Phi}^\top \boldsymbol{\Phi} \end{cases} \tag{8}$$

Secondly, if we consider the function $\forall s \in ]0, 1[ \quad \psi_{i,j}(s) = \frac{\gamma^2}{4} - \sum_{l=1}^d \mu_l^2 \frac{s(1-s)}{\beta_i^2 + s(\beta_j^2 \eta_l - \beta_i^2)}$ and $\boldsymbol{\mu} = \boldsymbol{\Phi}^\top (\hat{\boldsymbol{\theta}}_i - \hat{\boldsymbol{\theta}}_j)$ from Definition 5.1, we have the following:

$$
\begin{aligned}
\psi_{i,j}(s) &= \frac{\gamma^2}{4} - \sum_{l=1}^{d} \mu_l^2 \frac{s(1-s)}{\beta_i^2 + s(\beta_j^2 \eta_l - \beta_i^2)} \\
&= \frac{\gamma^2}{4} - \boldsymbol{\mu}^\top \mathrm{diag}(\frac{s(1-s)}{\beta_i^2 + s(\beta_j^2 \eta_l - \beta_i^2)}) \boldsymbol{\mu} \\
&= \frac{\gamma^2}{4} - \boldsymbol{\mu}^\top \mathrm{diag}(\frac{\beta_i^2}{s} + \frac{\beta_j^2}{1-s} \eta_l)^{-1} \boldsymbol{\mu} \\
&= \frac{\gamma^2}{4} - \boldsymbol{\mu}^\top \left( \frac{\beta_i^2}{s} \boldsymbol{I} + \frac{\beta_j^2}{1-s} \mathrm{diag}(\boldsymbol{\eta}) \right)^{-1} \boldsymbol{\mu} \\
&= \frac{\gamma^2}{4} - (\hat{\boldsymbol{\theta}}_i - \hat{\boldsymbol{\theta}}_j)^\top \left( \frac{\beta_i^2}{s} \boldsymbol{\Phi}\boldsymbol{\Phi}^\top + \frac{\beta_j^2}{1-s} \boldsymbol{\Phi}\mathrm{diag}(\boldsymbol{\eta})\boldsymbol{\Phi}^\top \right)^{-1} (\hat{\boldsymbol{\theta}}_i - \hat{\boldsymbol{\theta}}_j) \\
&= \frac{\gamma^2}{4} - (\hat{\boldsymbol{\theta}}_i - \hat{\boldsymbol{\theta}}_j)^\top \left( \frac{\beta_i^2}{s} \boldsymbol{A}_j^{-1} + \frac{\beta_j^2}{1-s} \boldsymbol{A}_i^{-1} \right)^{-1} (\hat{\boldsymbol{\theta}}_i - \hat{\boldsymbol{\theta}}_j) \quad \text{from Equation (8)} \\
&= \frac{\gamma^2}{4} - \|\hat{\boldsymbol{\theta}}_i - \hat{\boldsymbol{\theta}}_j\|^2_{\left( \frac{\beta_i^2}{1-s} \boldsymbol{A}_i^{-1} + \frac{\beta_j^2}{s} \boldsymbol{A}_j^{-1} \right)^{-1}}
\end{aligned}
$$

For $\gamma = 2$, we recover the function $\kappa$ as defined in Equation (4), from Gilitschenski & Hanebeck (2012), and conclude the property.

$\square$

## J. Appendix: Lower bound on the separation time

With our algorithm, once two agents are separated at iteration $t = T_s(i,j)$ (*i.e.,* $\Psi(i,j,t) = 1$), they no longer share samples. Hence the separation is definitive. We proof the lower bound on the separation time based on the parameter gap.

**Theorem 5.1** (Lower Bound on the Separation Time $T_s$ Between Two Agents). *Let $0 < \delta < 1$. Consider two agents $i$ and $j$, and suppose that Assumption 4.1 and Assumption 4.2 hold. Then, with probability at least $1 - \delta$, the separation time satisfies:*

$$
T_s(i,j) \geq \left\lceil \frac{8 \left( \mathrm{erf}^{-1}(1-\delta) - \mathrm{erf}(\delta) \right)^2}{\|\boldsymbol{\theta}_i^* - \boldsymbol{\theta}_j^*\|_2} \right\rceil ,
$$

*where $\mathrm{erf}(\cdot)$ denotes the Gauss error function.*

*Proof of Theorem 5.1.* To derive a lower bound, we examine a simplified case where the separation time occurs earlier than in any typical scenario.

We consider two agents $i$ and $j$, an iteration $T_s(i,j) \leq t$, we assume the nature of the noise is known and modeled as normalized Gaussian noise. We suppose that the set of arm is reduce to a single arm, *i.e.,* $\mathcal{X} = \left\{ \frac{\boldsymbol{\theta}_i^* - \boldsymbol{\theta}_j^*}{\|\boldsymbol{\theta}_i^* - \boldsymbol{\theta}_j^*\|_2} \right\}$, which the most favorable case to distinguish the two bandit parameters.

First, we bound $\|\hat{\boldsymbol{\theta}}_t - \boldsymbol{\theta}_i^*\|_{\boldsymbol{A}_t}$ in our specific setting:

$$
\begin{aligned}
\|\hat{\boldsymbol{\theta}}_t - \boldsymbol{\theta}_i^*\|_{\boldsymbol{A}_t} &= \|\boldsymbol{A}_t^{\frac{1}{2}} (\hat{\boldsymbol{\theta}}_t - \boldsymbol{\theta}_i^*)\|_2 \\
&= \|\frac{1}{\sqrt{t}} \boldsymbol{A}_t (\hat{\boldsymbol{\theta}}_t - \boldsymbol{\theta}_i^*)\|_2 \quad \text{since} \quad \boldsymbol{A}_t^{\frac{1}{2}} = \frac{1}{\sqrt{t}} \boldsymbol{A}_t = \sqrt{t} \boldsymbol{x}\boldsymbol{x}^\top \\
&= \left| \frac{1}{\sqrt{t}} \sum_{s=1}^{t} \eta_s \right| \quad \text{with} \quad \frac{1}{\sqrt{t}} \sum_{s=1}^{t} \eta_s \sim \mathcal{N}(0,1)
\end{aligned}
$$

We have $\mathbf{P}\left[\frac{1}{\sqrt{t}}\sum_{s=1}^{t}\eta_s \geq a\right] = 1 - \text{erf}(\frac{a}{\sqrt{2}})$. So with probability $1 - \delta$, we have:

$$\|\hat{\boldsymbol{\theta}}_t - \boldsymbol{\theta}_i^*\|_{\boldsymbol{A}_t} \leq \beta(\delta) \quad \text{with} \quad \beta(\delta) = \sqrt{2}\text{erf}^{-1}(1-\delta) \ .$$

Secondly, we bound $\|\hat{\boldsymbol{\theta}}_{i,t} - \hat{\boldsymbol{\theta}}_{j,t}\|_{\boldsymbol{A}_t}$ in our setting similarly to the previous approach, we have:

$$\|\hat{\boldsymbol{\theta}}_{i,t} - \hat{\boldsymbol{\theta}}_{j,t}\|_{\boldsymbol{A}_t} = \left\|\left(\sqrt{t}\|\boldsymbol{\theta}_i^* - \boldsymbol{\theta}_j^*\|_2 + \frac{1}{\sqrt{t}}\sum_{s=1}^{t}(\eta_{i,s} - \eta_{j,s})\right)\boldsymbol{x}\right\|_2$$

$$= |Z_{i,j,t}| \quad \text{with} \quad Z_{i,j,t} \sim \mathcal{N}(\sqrt{t}\|\boldsymbol{\theta}_i^* - \boldsymbol{\theta}_j^*\|_2, 2)$$

Lastly we consider our collaboration criterion with $t = T_s(i,j)$, we have:

$$\|\hat{\boldsymbol{\theta}}_{i,t} - \hat{\boldsymbol{\theta}}_{j,t}\|_{\boldsymbol{A}_t} \geq 2\beta(\delta)$$
$$|Z_{i,j,t}| \geq 2\beta(\delta)$$

Moreover setting $\mathbf{P}\left[|Z_{i,j,t}| \geq 2\beta(\delta)\right] = 1 - \delta$ and since $\mathbf{P}\left[|Z_{i,j,t}| \geq 2\beta(\delta)\right] = 2(1 - \mathbf{P}\left[Z_{i,j,t} \leq 2\beta(\delta)\right])$, we have:

$$1 - \delta = 2\left(1 - \left(\frac{1}{2} + \frac{1}{2}\text{erf}\left(\frac{1}{2\sqrt{2}}\left(2\beta(\delta) - \sqrt{t}\|\boldsymbol{\theta}_i^* - \boldsymbol{\theta}_j^*\|_2\right)\right)\right)\right)$$

$$1 - \delta = 1 - \text{erf}\left(\frac{1}{2\sqrt{2}}\left(2\beta(\delta) - \sqrt{t}\|\boldsymbol{\theta}_i^* - \boldsymbol{\theta}_j^*\|_2\right)\right)$$

$$2\sqrt{2}\text{erf}^{-1}(\delta) = 2\beta(\delta) - \sqrt{t}\|\boldsymbol{\theta}_i^* - \boldsymbol{\theta}_j^*\|_2$$

So, at iteration $t = T_s(i,j) = \frac{4(\beta(\delta) - \sqrt{2}\text{erf}^{-1}(\delta))^2}{\|\boldsymbol{\theta}_i^* - \boldsymbol{\theta}_j^*\|_2}$ with probability $1 - \delta$, we have $\|\hat{\boldsymbol{\theta}}_{i,t} - \hat{\boldsymbol{\theta}}_{j,t}\|_{\boldsymbol{A}_t} \geq 2\beta(\delta)$, which leads to:

$$T_s(i,j) \geq \left\lceil\frac{8(\text{erf}^{-1}(1-\delta) - \text{erf}(\delta))^2}{\|\boldsymbol{\theta}_i^* - \boldsymbol{\theta}_j^*\|_2}\right\rceil$$

Although we focus on a simple case, we demonstrate that the separation time scales inversely with $\|\boldsymbol{\theta}_i^* - \boldsymbol{\theta}_j^*\|_2$ which characterize how difficult the problem is.

$\square$

## K. Appendix: Upper bound on the separation time

**Theorem 6.1** (Upper Bound on the Separation Time $T_s$). *Let $0 < \delta < 1$, and consider two agents $i$ and $j$. Let $\epsilon_e > 0$ satisfy*

$$\epsilon_e \geq \frac{(1 + \sqrt{2})^2 R^2}{Q_{i,j}}, \text{ with } Q_{i,j} = \frac{1}{K}\sum_{k=1}^{K}\left(\boldsymbol{x}_k^\top(\boldsymbol{\theta}_i^* - \boldsymbol{\theta}_j^*)\right)^2 \ .$$

*Then, under Assumption 4.1, Assumption 4.2, and following Figure 1, we have, with probability at least $1 - \delta$,*

$$T_s(i,j) \leq \left\lceil\frac{4(2 + \sqrt{2})^2 R^2 \log\frac{1}{\delta}}{2\epsilon_e Q_{i,j} - (2 + \sqrt{2})^2 R^2}\right\rceil \ .$$

*Proof of Theorem 6.1.* Given two agents $i$ and $j$, at iteration $t \leq T_s(i,j)$, from Gilitschenski & Hanebeck (2012), under a

synchronous pulling, we have:

$$\|\hat{\boldsymbol{\theta}}_{i,t} - \hat{\boldsymbol{\theta}}_{j,t}\|_{\boldsymbol{A}_t} \leq 2\beta(\delta, \boldsymbol{A}_t)$$

$$\|\boldsymbol{\theta}_i^* - \boldsymbol{\theta}_j^*\|_{\boldsymbol{A}_t} - \|\boldsymbol{Z}_{i,j,t}\|_{\boldsymbol{A}_t^{-1}} \leq 2\beta(\delta, \boldsymbol{A}_t) \quad \text{from Lemma D.1}$$

$$\|\boldsymbol{\theta}_i^* - \boldsymbol{\theta}_j^*\|_{\boldsymbol{A}_t}^2 \leq (2+\sqrt{2})^2 \beta(\delta, \boldsymbol{A}_t)^2$$

$$\sum_{k=1}^{K} T_{kt}(\boldsymbol{x}_k^\top(\boldsymbol{\theta}_i^* - \boldsymbol{\theta}_j^*))^2 \leq (2+\sqrt{2})^2 \beta(\delta, \boldsymbol{A}_t)^2 \quad \text{with } T_{kt} \text{ the number of pulling of arm } \boldsymbol{x}_k \text{ at iteration } t$$

We have $T_{kt} = \epsilon_e T_{kt}^{\text{Unif}} + (1 - \epsilon_e)T_{kt}^{\text{UCB}}$, with $T_{kt}^{\text{Unif}}$ the number of pulling of arm $\boldsymbol{x}_k$ at iteration $t$ following the *Uniform* policy[4] and $T_{kt}^{\text{UCB}}$ the number of pulling of arm $\boldsymbol{x}_k$ at iteration $t$ following the *UCB* policy as depicted in Figure 1, we have with $t$ large enough:

$$T_{kt} = \epsilon_e T_{kt}^{\text{Unif}} + (1 - \epsilon_e)T_{kt}^{\text{UCB}} \geq \epsilon_e T_{kt}^{\text{Unif}} = \epsilon_e \frac{t}{K}$$

Thus, we have:

$$\epsilon_e \frac{t}{K} \sum_{k=1}^{K}(\boldsymbol{x}_k^\top(\boldsymbol{\theta}_i^* - \boldsymbol{\theta}_j^*))^2 \leq (2+\sqrt{2})^2 \beta(\delta, \boldsymbol{A}_t)^2$$

$$\epsilon_e \frac{t}{K} \sum_{k=1}^{K}(\boldsymbol{x}_k^\top(\boldsymbol{\theta}_i^* - \boldsymbol{\theta}_j^*))^2 \leq (2+\sqrt{2})^2 R^2 \left(2\log\frac{1}{\delta} + \log\det(\boldsymbol{A}_t)\right)$$

$$\epsilon_e \frac{t}{K} \sum_{k=1}^{K}(\boldsymbol{x}_k^\top(\boldsymbol{\theta}_i^* - \boldsymbol{\theta}_j^*))^2 \leq (2+\sqrt{2})^2 R^2 \left(2\log\frac{1}{\delta} + \frac{d}{2}\log\left(1 + \frac{t}{d}\right)\right)$$

$$\epsilon_e \frac{t}{K} \sum_{k=1}^{K}(\boldsymbol{x}_k^\top(\boldsymbol{\theta}_i^* - \boldsymbol{\theta}_j^*))^2 \leq (2+\sqrt{2})^2 R^2 \left(2\log\frac{1}{\delta} + \frac{t}{2}\right)$$

$$t\left(\frac{\epsilon_e}{K}\sum_{k=1}^{K}(\boldsymbol{x}_k^\top(\boldsymbol{\theta}_i^* - \boldsymbol{\theta}_j^*))^2 - \frac{(2+\sqrt{2})^2 R^2}{2}\right) \leq 2(2+\sqrt{2})^2 R^2 \log\frac{1}{\delta}$$

Finally, we conclude that:

$$T_s(i,j) \leq \left\lceil \frac{4(2+\sqrt{2})^2 R^2 \log\frac{1}{\delta}}{2\epsilon_e Q_{i,j} - (2+\sqrt{2})^2 R^2} \right\rceil$$

with an Signal to Noise Ratio (SNR) such as: $\frac{\sqrt{Q_{i,j}}}{R} \geq \frac{(2+\sqrt{2})}{\sqrt{2\epsilon_e}}$ and $Q_{i,j} = \frac{1}{K}\sum_{k=1}^{K}(\boldsymbol{x}_k^\top(\boldsymbol{\theta}_i^* - \boldsymbol{\theta}_j^*))^2$

$\square$

## L. Appendix: Cumulative pseudo-regret upper bound

**Theorem 6.2** (Individual Regret During the Collaboration Phase). *Let $T_s = T_s(i,j)$ denote the separation time introduced in Definition 5.2. Under Assumption 4.1 and Assumption 4.2, the cumulative regret of agent $i$ during the collaboration phase is bounded as:*

$$R_{i,0,T_s}^{\text{collab}} \leq \mu(\delta, d, \gamma) \cdot \nu(\delta, d, T_s) + \frac{T_s}{2}\left(\boldsymbol{\theta}_i^* - \boldsymbol{\theta}_j^*\right)^\top \boldsymbol{x}_i^* \;,$$

*where*

$$\mu(\delta, d, \gamma) = \frac{1}{2} + \frac{\gamma}{4} + \frac{1}{2\sqrt{2}}\sqrt{1 + \frac{d\log 2}{2\log\frac{1}{\delta}}} \;,$$

---

[4]This highlight the particular utility of the $\epsilon-$uniform sampling , as the typical UCB lower bounds are asymptotically.

*and*

$$\nu(\delta, d, T_s) = \sqrt{4\beta(\delta, \boldsymbol{A}_{T_s})^2 T_s d \log\left(1 + \frac{T_s}{d}\right)} \ .$$

*Proof of* Theorem 6.2. We consider the case of two agents $i$ and $j$, focusing the instantaneous regret $r_{i,t}$, given that $t \leq T_s(i,j)$ we have $\mathcal{N}_i(t) = \{j\}$, from Equation (5), we have:

$$
\begin{aligned}
r_{i,t} &\leq \left(\tilde{\boldsymbol{\theta}}_{\widehat{\mathcal{N}}_i(t)} - \boldsymbol{\theta}^*_{\widehat{\mathcal{N}}_i(t)} - \boldsymbol{\theta}^*_i + \boldsymbol{\theta}^*_{\widehat{\mathcal{N}}_i(t)}\right)^\top \boldsymbol{x}_{\widehat{\mathcal{N}}_i(t)} + (\boldsymbol{\theta}^*_i - \boldsymbol{\theta}^*_{\widehat{\mathcal{N}}_i(t)})^\top \boldsymbol{x}^*_i \\
&\leq \left(\frac{\beta(\delta, \boldsymbol{A}_{\widehat{\mathcal{N}}_i(t)})}{\|\boldsymbol{x}_{\widehat{\mathcal{N}}_i(t)}\|_{\boldsymbol{A}^{-1}_{\widehat{\mathcal{N}}_i(t)}}} \boldsymbol{A}^{-1\top}_{\widehat{\mathcal{N}}_i(t)} \boldsymbol{x}_{\widehat{\mathcal{N}}_i(t)} + \widehat{\boldsymbol{\theta}}_{\widehat{\mathcal{N}}_i(t)} - \boldsymbol{\theta}^*_i\right)^\top \boldsymbol{x}_{\widehat{\mathcal{N}}_i(t)} + \frac{1}{2}(\boldsymbol{\theta}^*_i - \boldsymbol{\theta}^*_j)^\top \boldsymbol{x}^*_i \\
&\leq \beta(\delta, \boldsymbol{A}_{\widehat{\mathcal{N}}_i(t)})\|\boldsymbol{x}_{\widehat{\mathcal{N}}_i(t)}\|_{\boldsymbol{A}^{-1}_{\widehat{\mathcal{N}}_i(t)}} + \left(\frac{1}{2}(\widehat{\boldsymbol{\theta}}_i + \widehat{\boldsymbol{\theta}}_j) - \boldsymbol{\theta}^*_i + \frac{1}{2}(\widehat{\boldsymbol{\theta}}_i - \widehat{\boldsymbol{\theta}}_i)\right)^\top \boldsymbol{x}_{\widehat{\mathcal{N}}_i(t)} + \frac{1}{2}(\boldsymbol{\theta}^*_i - \boldsymbol{\theta}^*_j)^\top \boldsymbol{x}^*_i \\
&\leq \frac{1}{\sqrt{2}}\beta(\delta, 2\boldsymbol{A}_t)\|\boldsymbol{x}_{\widehat{\mathcal{N}}_i(t)}\|_{\boldsymbol{A}^{-1}_t} + \left(\frac{1}{2}\|\widehat{\boldsymbol{\theta}}_i - \widehat{\boldsymbol{\theta}}_j\|_{\boldsymbol{A}_t} + \|\widehat{\boldsymbol{\theta}}_i - \boldsymbol{\theta}^*_i\|_{\boldsymbol{A}_t}\right)\|\boldsymbol{x}_{\widehat{\mathcal{N}}_i(t)}\|_{\boldsymbol{A}^{-1}_t} + \frac{1}{2}(\boldsymbol{\theta}^*_i - \boldsymbol{\theta}^*_j)^\top \boldsymbol{x}^*_i \\
&\leq \left(\frac{1}{\sqrt{2}}\beta(\delta, 2\boldsymbol{A}_t) + \frac{\gamma}{2}\beta(\delta, \boldsymbol{A}_t) + \beta(\delta, \boldsymbol{A}_t)\right)\|\boldsymbol{x}_{\widehat{\mathcal{N}}_i(t)}\|_{\boldsymbol{A}^{-1}_t} + \frac{1}{2}(\boldsymbol{\theta}^*_i - \boldsymbol{\theta}^*_j)^\top \boldsymbol{x}^*_i \\
&\leq 2\left(\frac{1}{2\sqrt{2}}\frac{\beta(\delta, 2\boldsymbol{A}_t)}{\beta(\delta, \boldsymbol{A}_t)} + \frac{\gamma}{4} + \frac{1}{2}\right)\beta(\delta, \boldsymbol{A}_t)\|\boldsymbol{x}_{\widehat{\mathcal{N}}_i(t)}\|_{\boldsymbol{A}^{-1}_t} + \frac{1}{2}(\boldsymbol{\theta}^*_i - \boldsymbol{\theta}^*_j)^\top \boldsymbol{x}^*_i \\
&\leq 2\left(\frac{1}{2} + \frac{\gamma}{4} + \frac{1}{2\sqrt{2}}\frac{\beta(\delta, 2\boldsymbol{A}_t)}{\beta(\delta, \boldsymbol{A}_t)}\right)\beta(\delta, \boldsymbol{A}_t)\|\boldsymbol{x}_{\widehat{\mathcal{N}}_i(t)}\|_{\boldsymbol{A}^{-1}_t} + \frac{1}{2}(\boldsymbol{\theta}^*_i - \boldsymbol{\theta}^*_j)^\top \boldsymbol{x}^*_i \\
&\leq 2\left(\frac{1}{2} + \frac{\gamma}{4} + \frac{1}{2\sqrt{2}}\sqrt{1 + \frac{d\log 2}{2\log\frac{1}{\delta}}}\right)\beta(\delta, \boldsymbol{A}_t)\|\boldsymbol{x}_{\widehat{\mathcal{N}}_i(t)}\|_{\boldsymbol{A}^{-1}_t} + \frac{1}{2}(\boldsymbol{\theta}^*_i - \boldsymbol{\theta}^*_j)^\top \boldsymbol{x}^*_i \quad \text{from Lemma D.4} \\
&\leq 2\mu(\delta, d, \gamma)\beta(\delta, \boldsymbol{A}_t)\|\boldsymbol{x}_{\widehat{\mathcal{N}}_i(t)}\|_{\boldsymbol{A}^{-1}_t} + \frac{1}{2}(\boldsymbol{\theta}^*_i - \boldsymbol{\theta}^*_j)^\top \boldsymbol{x}^*_i \quad \text{with} \quad \mu(\delta, d, \gamma) = \frac{1}{2} + \frac{\gamma}{4} + \frac{1}{2\sqrt{2}}\sqrt{1 + \frac{d\log 2}{2\log\frac{1}{\delta}}}
\end{aligned}
$$

With Abbasi-Yadkori et al. (2011) and by setting $\nu(\delta, T_s) = \sqrt{4\beta(\delta, \boldsymbol{A}_{T_s})^2 T_s d \log(1 + \frac{T_s}{d})}$, we have:

$$R^{\text{collab}}_{i,0,T_s} \leq \mu(\delta, d, \gamma)\nu(\delta, T_s) + \frac{T_s}{2}(\boldsymbol{\theta}^*_i - \boldsymbol{\theta}^*_j)^\top \boldsymbol{x}^*_i \ .$$

$\square$

Note that in order to enforce $\mu(\delta, d, \gamma) < 1$, we need to set $\gamma < 2 - \sqrt{2}\sqrt{1 + (d\log 2)/(2\log\frac{1}{\delta})}$. For example, for $d = 10$ and $\delta = 0.001$, we have is $\gamma \leq 0.27$.

Moreover, recall that in Gilitschenski & Hanebeck (2012), the authors consider $\gamma = 2$ to derive their ellipsoid separation test. Here, the term $\sqrt{1 + (d\log 2)/(2\log\frac{1}{\delta})}$ could be interpreted as a correction that accounts for the regret minimization objective. Note that this term is derived from the loose upper-bound of $\beta(\delta, 2\boldsymbol{A}_t)/\beta(\delta, \boldsymbol{A}_t)$, which is close to 1 and relatively insensitive to both the number of agents and the problem's dimensionality.

## M. Appendix: Network cumulative pseudo-regret upper bound

**Theorem 6.3** (Regret During the Collaboration Phase). *Let $T_s = T_s(i,j)$ denote the separation time introduced in Definition 5.2. Under Assumption 4.1 and Assumption 4.2, the cumulative regret of all agents during the collaboration phase*

*is bounded as:*

$$\bar{R}_{0,T_s}^{\text{collab}} \le \mu'(\delta, d) \cdot \nu(\delta, d, T_s) + \frac{T_s}{4} \left(\boldsymbol{\theta}_i^* - \boldsymbol{\theta}_j^*\right)^\top \left(\boldsymbol{x}_i^* - \boldsymbol{x}_j^*\right) \ ,$$

*where*

$$\mu'(\delta, d) = \frac{1}{\sqrt{2}} \sqrt{1 + \frac{d \log 2}{2 \log \frac{1}{\delta}}} \ ,$$

*and $\nu(\delta, d, T_s)$ is as previously defined.*

*Proof of* Theorem 6.3. We consider the case of two agents $i$ and $j$, focusing the averaged instantaneous regret $\bar{r}_{i,t} = \frac{1}{2}(r_{i,t} + r_{j,t})$, given that $t \le T_s(i,j)$ we have $\mathcal{N}_i(t) = \{j\}$ and $\mathcal{N}_j(t) = \{i\}$, which give us:

$$\begin{aligned}
\bar{r}_{i,t} &= \frac{1}{2}(r_{i,t} + r_{j,t}) \\
&= \frac{1}{2}(\boldsymbol{\theta}_i^{*\top}(\boldsymbol{x}_i^* - \boldsymbol{x}_{\widehat{\mathcal{N}}_i(t)}) + \boldsymbol{\theta}_j^{*\top}(\boldsymbol{x}_j^* - \boldsymbol{x}_{\widehat{\mathcal{N}}_i(t)})) \\
&= -\boldsymbol{\theta}_{\widehat{\mathcal{N}}_i(t)}^{*\top} \boldsymbol{x}_{\widehat{\mathcal{N}}_i(t)} + \frac{1}{2}\boldsymbol{\theta}_i^{*\top}\boldsymbol{x}_i^* + \frac{1}{2}\boldsymbol{\theta}_j^{*\top}\boldsymbol{x}_j^* \\
&= \boldsymbol{\theta}_{\widehat{\mathcal{N}}_i(t)}^{*\top}(\boldsymbol{x}_{\widehat{\mathcal{N}}_i(t)}^* - \boldsymbol{x}_{\widehat{\mathcal{N}}_i(t)}) - \boldsymbol{\theta}_{\widehat{\mathcal{N}}_i(t)}^{*\top}\boldsymbol{x}_{\widehat{\mathcal{N}}_i(t)}^* + \frac{1}{2}\boldsymbol{\theta}_i^{*\top}\boldsymbol{x}_i^* + \frac{1}{2}\boldsymbol{\theta}_j^{*\top}\boldsymbol{x}_j^*
\end{aligned}$$

Moreover, since $\boldsymbol{\theta}_{\widehat{\mathcal{N}}_i(t)}^{*\top}\boldsymbol{x}_{\widehat{\mathcal{N}}_i(t)}^* \ge \boldsymbol{\theta}_{\widehat{\mathcal{N}}_i(t)}^{*\top}\left(\frac{1}{2}(\boldsymbol{x}_i^* + \boldsymbol{x}_j^*)\right)$, we have:

$$\begin{aligned}
\bar{r}_{i,t} &\le \boldsymbol{\theta}_{\widehat{\mathcal{N}}_i(t)}^{*\top}(\boldsymbol{x}_{\widehat{\mathcal{N}}_i(t)}^* - \boldsymbol{x}_{\widehat{\mathcal{N}}_i(t)}) - \boldsymbol{\theta}_{\widehat{\mathcal{N}}_i(t)}^{*\top}\left(\frac{1}{2}(\boldsymbol{x}_i^* + \boldsymbol{x}_j^*)\right) + \frac{1}{2}\boldsymbol{\theta}_i^{*\top}\boldsymbol{x}_i^* + \frac{1}{2}\boldsymbol{\theta}_j^{*\top}\boldsymbol{x}_j^* \\
&\le \boldsymbol{\theta}_{\widehat{\mathcal{N}}_i(t)}^{*\top}(\boldsymbol{x}_{\widehat{\mathcal{N}}_i(t)}^* - \boldsymbol{x}_{\widehat{\mathcal{N}}_i(t)}) + \frac{1}{2}\left(-\frac{1}{2}\boldsymbol{\theta}_i^{*\top}\boldsymbol{x}_i^* - \frac{1}{2}\boldsymbol{\theta}_i^{*\top}\boldsymbol{x}_j^* - \frac{1}{2}\boldsymbol{\theta}_j^{*\top}\boldsymbol{x}_i^* - \frac{1}{2}\boldsymbol{\theta}_j^{*\top}\boldsymbol{x}_j^* + \boldsymbol{\theta}_i^{*\top}\boldsymbol{x}_i^* + \boldsymbol{\theta}_j^{*\top}\boldsymbol{x}_j^*\right) \\
&\le \boldsymbol{\theta}_{\widehat{\mathcal{N}}_i(t)}^{*\top}(\boldsymbol{x}_{\widehat{\mathcal{N}}_i(t)}^* - \boldsymbol{x}_{\widehat{\mathcal{N}}_i(t)}) + \frac{1}{4}(\boldsymbol{\theta}_i^* - \boldsymbol{\theta}_j^*)^\top(\boldsymbol{x}_i^* - \boldsymbol{x}_j^*) \\
&\le 2\beta(\delta, \boldsymbol{A}_{\widehat{\mathcal{N}}_i(t)})\|\boldsymbol{x}_{\widehat{\mathcal{N}}_i(t)}\|_{\boldsymbol{A}_{\widehat{\mathcal{N}}_i(t)}^{-1}} + \frac{1}{4}(\boldsymbol{\theta}_i^* - \boldsymbol{\theta}_j^*)^\top(\boldsymbol{x}_i^* - \boldsymbol{x}_j^*) \\
&\le \sqrt{2}\beta(\delta, 2\boldsymbol{A}_t)\|\boldsymbol{x}_{\widehat{\mathcal{N}}_i(t)}\|_{\boldsymbol{A}_t^{-1}} + \frac{1}{4}(\boldsymbol{\theta}_i^* - \boldsymbol{\theta}_j^*)^\top(\boldsymbol{x}_i^* - \boldsymbol{x}_j^*) \\
&\le 2\left(\frac{1}{\sqrt{2}}\frac{\beta(\delta, 2\boldsymbol{A}_t)}{\beta(\delta, \boldsymbol{A}_t)}\right)\beta(\delta, \boldsymbol{A}_t)\|\boldsymbol{x}_{\widehat{\mathcal{N}}_i(t)}\|_{\boldsymbol{A}_t^{-1}} + \frac{1}{4}(\boldsymbol{\theta}_i^* - \boldsymbol{\theta}_j^*)^\top(\boldsymbol{x}_i^* - \boldsymbol{x}_j^*) \\
&\le 2\left(\frac{1}{\sqrt{2}}\sqrt{1 + \frac{d\log 2}{2\log\frac{1}{\delta}}}\right)\beta(\delta, \boldsymbol{A}_t)\|\boldsymbol{x}_{\widehat{\mathcal{N}}_i(t)}\|_{\boldsymbol{A}_t^{-1}} + \frac{1}{4}(\boldsymbol{\theta}_i^* - \boldsymbol{\theta}_j^*)^\top(\boldsymbol{x}_i^* - \boldsymbol{x}_j^*) \\
&\le 2\mu'(\delta, d)\beta(\delta, \boldsymbol{A}_t)\|\boldsymbol{x}_{\widehat{\mathcal{N}}_i(t)}\|_{\boldsymbol{A}_t^{-1}} + \frac{1}{4}(\boldsymbol{\theta}_i^* - \boldsymbol{\theta}_j^*)^\top(\boldsymbol{x}_i^* - \boldsymbol{x}_j^*) \quad \text{with} \quad \mu'(\delta, d) = \frac{1}{\sqrt{2}}\sqrt{1 + \frac{d\log 2}{2\log\frac{1}{\delta}}}
\end{aligned}$$

With Abbasi-Yadkori et al. (2011) and by setting $\nu(\delta, T_s) = \sqrt{4\beta(\delta, \boldsymbol{A}_{T_s})^2 T_s d \log(1 + \frac{T_s}{d})}$, we have:

$$\bar{R}_{0,T_s}^{\text{collab}} \le \mu'(\delta, d)\nu(\delta, T_s) + \frac{T_s}{4}(\boldsymbol{\theta}_i^* - \boldsymbol{\theta}_j^*)^\top(\boldsymbol{x}_i^* - \boldsymbol{x}_j^*) \ .$$

$\square$

Note that for reasonable values of $d$ and $\delta$, we have: $\mu'(\delta, d) \le \mu(\delta, d, \gamma)$ which gives us the intuition that the network of agents is an improvement over the single agent case.

The same intuition applies for the second term $\frac{T_s}{4}(\boldsymbol{\theta}_i^* - \boldsymbol{\theta}_j^*)^\top (\boldsymbol{x}_i^* - \boldsymbol{x}_j^*)$. Since the bandit parameters and the arms norms are bounded, the only configuration for which this inner product is high, is when at least one arm is approximately in the direction of $\boldsymbol{\theta}_i^* - \boldsymbol{\theta}_j^*$. With this conditioning, the separation of the two bandit parameters is accelerated and the collaboration ceases shortly after.

In summary, taking into account the averaged cumulative pseudo regret better underlines the *adaptive* aspect of our collaboration.

## N. Appendix: Expected number of misassigned agents

We detail the proof of the upper-bound on the clustering error in Lemma 6.1 .

**Lemma 6.1** (Expected Number of Misassigned Agents). *Let* $0 < \delta < 1$. *Under Assumption 4.1, Assumption 4.2, and Assumption 6.1, we have, with probability at least* $1 - \delta$:

$$\mathbb{E}\left[N_i^e(t)\right] \leq n_i^e(t) \ ,$$

*where*

$$n_i^e(t) = \left\lceil \frac{N\sqrt{\det(\boldsymbol{A}_t)}}{\exp\left(\frac{1}{2}\left(\Delta_{\boldsymbol{A}_t}^{\min} - \gamma\beta(\delta, \boldsymbol{A}_t)\right)^2\right)} - \rho_{m(i)}N \right\rceil \ ,$$

*and*

$$\Delta_{\boldsymbol{A}_t}^{\min} = \min_{i\neq j} \|\boldsymbol{\theta}_i^* - \boldsymbol{\theta}_j^*\|_{\boldsymbol{A}_t} \ .$$

*Proof of Lemma 6.1.* We have, at iteration $t > 0$, $N_{i,t}^e = |\hat{N}_{i,t} - \rho_{m(i)}N|$ with $\hat{N}_{i,t}$ the number of agents within the estimated cluster, we have:

$$\begin{aligned}
\mathbf{E}\left[N_{i,t}^e\right] &= \mathbf{E}\left[|\hat{N}_{i,t} - \rho_{m(i)}N|\right] \\
&= \mathbf{E}\left[\hat{N}_{i,t} - \rho_{m(i)}N\right] \quad \text{from Lemma D.3} \\
&= \mathbf{E}\left[\sum_{j=1}^{N} \mathbf{1}_{\{\Psi(i,j,t)=0\}} - \rho_{m(i)}N\right] \\
&= \sum_{j=1}^{N} \mathbf{P}\left[\Psi(i,j,t) = 0\right] - \rho_{m(i)}N
\end{aligned}$$

Focusing on $\mathbf{P}\left[\Psi(i,j,t) = 0\right]$, we have:

$$\begin{aligned}
\mathbf{P}\left[\Psi(i,j,t) = 0\right] &= \mathbf{P}\left[\|\hat{\boldsymbol{\theta}}_i - \hat{\boldsymbol{\theta}}_j\|_{\boldsymbol{A}_t} \leq 2\beta(\delta, \boldsymbol{A}_t)\right] \\
&\leq \mathbf{P}\left[\|\boldsymbol{Z}_{i,j,t}\|_{\boldsymbol{A}_t^{-1}} \geq 2\beta(\delta, \boldsymbol{A}_t) - \|\boldsymbol{\theta}_i^* - \boldsymbol{\theta}_j^*\|_{\boldsymbol{A}_t}\right]
\end{aligned}$$

By introducing $u > 0$ and by using the Chernoff bound, we have:

$$\begin{aligned}
\mathbf{P}\left[\Psi(i,j,t) = 0\right] &\leq \exp(-u) \quad \text{with } \sqrt{2u + \log \det \boldsymbol{A}_t} = \|\boldsymbol{\theta}_i^* - \boldsymbol{\theta}_j^*\|_{\boldsymbol{A}_t} - 2\beta(\delta, \boldsymbol{A}_t) \\
&\leq \exp\left(-\frac{1}{2}\left(\|\boldsymbol{\theta}_i^* - \boldsymbol{\theta}_j^*\|_{\boldsymbol{A}_t} - 2\beta(\delta, \boldsymbol{A}_t)\right)^2 + \frac{1}{2}\log\det\boldsymbol{A}_t\right) \\
\mathbf{P}\left[\Psi(i,j,t) = 0\right] &\leq \sqrt{\frac{\det(\boldsymbol{A}_t)}{\exp((\Delta_{\boldsymbol{\theta}^*,\boldsymbol{A}_t}^{\min} - \gamma\beta(\delta, \boldsymbol{A}_t))^2)}}
\end{aligned}$$

Which yield us the desired result:

$$\mathbb{E}\left[N_i^e(t)\right] \leq \left\lceil \left(\frac{N\sqrt{\det(\boldsymbol{A}_t)}}{\exp(\frac{1}{2}(\Delta_{\boldsymbol{\theta}^*,\boldsymbol{A}_t}^{\min} - \gamma\beta(\delta,\boldsymbol{A}_t))^2)} - \rho_{m(i)}N\right)\right\rceil$$

where $\Delta_{\boldsymbol{A}_t}^{\min} = \min\limits_{i,j \;\; i \neq j} \|\boldsymbol{\theta}_i^* - \boldsymbol{\theta}_j^*\|_{\boldsymbol{A}_t}$ and $\rho_{m(i)}N$ the number of agents in the cluster $m(i)$, the agent's cluster.

$\square$

## O. Appendix: Cumulative pseudo regret with clustered agents

**Theorem 6.4** (Cumulative Pseudo-Regret with Clustered Agents). *Under Assumption 4.1, Assumption 4.2, and Assumption 6.1, after $T$ iterations of the* BASS *algorithm, the cumulative pseudo-regret of agent $i$ satisfies:*

$$R_{i,0,T}^{\text{cluster}} \leq \mu''(\delta, d, N) \cdot \nu(\delta, d, T) + \frac{4L}{\delta \rho_{\min} N} \cdot C_i(T_c) \; ,$$

*where:*

$$C_i(T_c) = \sum_{t=1}^{\min(T,T_c)} n_i^e(t) \; ,$$

$$\mu''(\delta, d, N) = \frac{1}{\sqrt{\rho_{\min}N}} \sqrt{1 + \frac{d \log N}{2 \log \frac{1}{\delta}}} \; ,$$

$$T_c = \max_j T_s(i,j), \quad \rho_{\min} = \min_j \rho_{m(j)} \; ,$$

*and $\nu(\delta, d, T)$ is as previously defined.*

*Proof of Theorem 6.4.* At round $t$, for a given agent $i$, by considering the optimistic arm $\boldsymbol{x}_{\widehat{\mathcal{N}}_i(t)}$ and the optimal arm $\boldsymbol{x}_i^*$, from Equation (5), we have:

$$
\begin{aligned}
r_{i,t} &\le (\tilde{\boldsymbol{\theta}}_{\widehat{\mathcal{N}}_i(t)} - \boldsymbol{\theta}^*_{\widehat{\mathcal{N}}_i(t)})^\top \boldsymbol{x}_{\widehat{\mathcal{N}}_i(t)} + (\boldsymbol{\theta}^*_i - \boldsymbol{\theta}^*_{\widehat{\mathcal{N}}_i(t)})^\top (\boldsymbol{x}^*_i - \boldsymbol{x}_{\widehat{\mathcal{N}}_i(t)}) \\
&\le (\tilde{\boldsymbol{\theta}}_{\widehat{\mathcal{N}}_i(t)} - \boldsymbol{\theta}^*_{\widehat{\mathcal{N}}_i(t)})^\top \boldsymbol{x}_{\widehat{\mathcal{N}}_i(t)} + 2\|\boldsymbol{\theta}^*_{\widehat{\mathcal{N}}_i(t)} - \boldsymbol{\theta}^*_i\|_2 \quad \text{from Assumption 4.1} \\
&\le \left( \widehat{\boldsymbol{\theta}}_{\widehat{\mathcal{N}}_i(t)} + \frac{\beta(\delta, \boldsymbol{A}_{\widehat{\mathcal{N}}_i(t)})}{\|\boldsymbol{x}_{\widehat{\mathcal{N}}_i(t)}\|_{\boldsymbol{A}^{-1}_{\widehat{\mathcal{N}}_i(t)}}} \boldsymbol{A}^{-1\top}_{\widehat{\mathcal{N}}_i(t)} \boldsymbol{x}_{\widehat{\mathcal{N}}_i(t)} - \boldsymbol{\theta}^*_{\widehat{\mathcal{N}}_i(t)} \right)^\top \boldsymbol{x}_{\widehat{\mathcal{N}}_i(t)} + 2\|\boldsymbol{\theta}^*_{\widehat{\mathcal{N}}_i(t)} - \boldsymbol{\theta}^*_i\|_2 \quad \text{from Lemma D.2} \\
&\le \beta(\delta, \boldsymbol{A}_{\widehat{\mathcal{N}}_i(t)})\|\boldsymbol{x}_{\widehat{\mathcal{N}}_i(t)}\|_{\boldsymbol{A}^{-1}_{\widehat{\mathcal{N}}_i(t)}} + \left( \widehat{\boldsymbol{\theta}}_{\widehat{\mathcal{N}}_i(t)} - \boldsymbol{\theta}^*_{\widehat{\mathcal{N}}_i(t)} \right)^\top \boldsymbol{x}_{\widehat{\mathcal{N}}_i(t)} + 2\|\boldsymbol{\theta}^*_{\widehat{\mathcal{N}}_i(t)} - \boldsymbol{\theta}^*_i\|_2 \\
&\le \left( \beta(\delta, \boldsymbol{A}_{\widehat{\mathcal{N}}_i(t)}) + \|\widehat{\boldsymbol{\theta}}_{\widehat{\mathcal{N}}_i(t)} - \boldsymbol{\theta}^*_{\widehat{\mathcal{N}}_i(t)}\|_{\boldsymbol{A}_{\widehat{\mathcal{N}}_i(t)}} \right) \|\boldsymbol{x}_{\widehat{\mathcal{N}}_i(t)}\|_{\boldsymbol{A}^{-1}_{\widehat{\mathcal{N}}_i(t)}} + 2\|\boldsymbol{\theta}^*_{\widehat{\mathcal{N}}_i(t)} - \boldsymbol{\theta}^*_i\|_2 \\
&\le 2\beta(\delta, \boldsymbol{A}_{\widehat{\mathcal{N}}_i(t)})\|\boldsymbol{x}_{\widehat{\mathcal{N}}_i(t)}\|_{\boldsymbol{A}^{-1}_{\widehat{\mathcal{N}}_i(t)}} + 2\|\boldsymbol{\theta}^*_{\widehat{\mathcal{N}}_i(t)} - \boldsymbol{\theta}^*_i\|_2 \\
&\le \frac{2}{\sqrt{N_i + N^e_i(t)}}\beta(\delta, (N_i + N^e_i(t))\boldsymbol{A}_t)\|\boldsymbol{x}_{\widehat{\mathcal{N}}_i(t)}\|_{\boldsymbol{A}^{-1}_t} + 2\|\boldsymbol{\theta}^*_{\widehat{\mathcal{N}}_i(t)} - \boldsymbol{\theta}^*_i\|_2 \\
&\le \frac{2}{\sqrt{N_i + N^e_i(t)}}\beta(\delta, (N_i + N^e_i(t))\boldsymbol{A}_t)\|\boldsymbol{x}_{\widehat{\mathcal{N}}_i(t)}\|_{\boldsymbol{A}^{-1}_t} + 2\left\| \frac{-N^e_i(t)}{N_i + N^e_i(t)}\boldsymbol{\theta}^*_i + \frac{1}{N_i + N^e_i(t)}\sum_{j \in \widehat{\mathcal{N}}_i \backslash \mathcal{N}_i} \boldsymbol{\theta}^*_j \right\|_2 \\
&\le \frac{2}{\sqrt{N_i + N^e_i(t)}}\beta(\delta, (N_i + N^e_i(t))\boldsymbol{A}_t)\|\boldsymbol{x}_{\widehat{\mathcal{N}}_i(t)}\|_{\boldsymbol{A}^{-1}_t} + 2\frac{N^e_i(t)}{N_i + N^e_i(t)}L + \frac{1}{N_i + N^e_i(t)}\sum_{j \in \widehat{\mathcal{N}}_i \backslash \mathcal{N}_i} L \quad \text{from Assumption 4.1} \\
&\le \frac{2}{\sqrt{N_i + N^e_i(t)}}\beta(\delta, (N_i + N^e_i(t))\boldsymbol{A}_t)\|\boldsymbol{x}_{\widehat{\mathcal{N}}_i(t)}\|_{\boldsymbol{A}^{-1}_t} + 2\frac{2LN^e_i(t)}{N_i + N^e_i(t)} \\
&\le \frac{2}{\sqrt{N_i + N^e_i(t)}}\beta(\delta, (N_i + N^e_i(t))\boldsymbol{A}_t)\|\boldsymbol{x}_{\widehat{\mathcal{N}}_i(t)}\|_{\boldsymbol{A}^{-1}_t} + \frac{4L}{\delta N_i}n^e_i(t) \quad \text{from Lemma 6.1 and Markov inequality} \\
&\le \frac{2}{\sqrt{\rho_{\min}N}}\beta(\delta, N\boldsymbol{A}_t)\|\boldsymbol{x}_{\widehat{\mathcal{N}}_i(t)}\|_{\boldsymbol{A}^{-1}_t} + \frac{4L}{\delta\rho_{\min}N}n^e_i(t) \\
&\le 2\left( \frac{1}{\sqrt{\rho_{\min}N}}\frac{\beta(\delta, N\boldsymbol{A}_t)}{\beta(\delta, \boldsymbol{A}_t)} \right)\beta(\delta, \boldsymbol{A}_t)\|\boldsymbol{x}_{\widehat{\mathcal{N}}_i(t)}\|_{\boldsymbol{A}^{-1}_t} + \frac{4L}{\delta\rho_{\min}N}n^e_i(t) \\
&\le 2\left( \frac{1}{\sqrt{\rho_{\min}N}}\sqrt{1 + \frac{d\log N}{2\log\frac{1}{\delta}}} \right)\beta(\delta, \boldsymbol{A}_t)\|\boldsymbol{x}_{\widehat{\mathcal{N}}_i(t)}\|_{\boldsymbol{A}^{-1}_t} + \frac{4L}{\delta\rho_{\min}N}n^e_i(t) \\
&\le 2\mu''(\delta, d, N)\beta(\delta, \boldsymbol{A}_t)\|\boldsymbol{x}_{\widehat{\mathcal{N}}_i(t)}\|_{\boldsymbol{A}^{-1}_t} + \frac{4L}{\delta\rho_{\min}N}n^e_i(t) \quad \text{with} \quad \mu''(\delta, d, N) = \frac{1}{\sqrt{\rho_{\min}N}}\sqrt{1 + \frac{d\log N}{2\log\frac{1}{\delta}}}
\end{aligned}
$$

With Abbasi-Yadkori et al. (2011) and by setting $\nu(\delta, T_s) = \sqrt{4\beta(\delta, \boldsymbol{A}_{T_s})^2 T_s d\log(1 + \frac{T_s}{d})}$, we have:

$$
R^{\text{cluster}}_{i,0,T} \le \mu''(\delta, d, N)\nu(\delta, T_s) + \frac{4L}{\delta\rho_{\min}N}C_i(T_c) ,
$$

with $C_i(T_c) = \sum_{t=1}^{\min(T,T_c)} n^e_i(t)$, $n^e_i(t)$ defined in Lemma 6.1, $Tc = \max_j T_s(i, j)$ and $\rho_{\min} = \min_j \rho_{m(j)}$.

Note that this upper bound is somewhat loose, as it introduces the iteration $T_c$ at which the cluster is adequately estimated. While it does not highlight the *adaptive* nature of our collaboration, it still provides the asymptotic behavior of the *BASS* algorithm.

$\square$

# P. Appendix: Additional theoretical analysis

We gather additional theoretical results here to provide comparative baselines of our analysis.

## P.1. Proof of the cumulative pseudo regret upper-bound for the *BASS-Oracle* algorithm

**Corollary P.1** (Cumulative pseudo regret upper-bound for *Oracle*). *Under the assumptions made above, if we consider a* Oracle *controller that know in advance the true clusters, after $T$ iterations the pseudo-regret can be bounded as follows:*

$$R_{i,0,T}^{\text{oracle}} \leq \mu''(\delta, T)\nu(\delta, T) \ ,$$

*with $\mu''(\delta, d) = \frac{1}{\sqrt{N}}\sqrt{1 + \frac{d\log N}{2\log\frac{1}{\delta}}}$ and $\nu(\delta, T)$ defined as previously,*

*Proof of* Corollary P.1. This is a direct application of Theorem 6.4 with a null neighborhood estimation error *i.e.*, for $i$ and $t > 0$, we have $N_i^e(t) = 0$.

$\square$

## P.2. Asymptotic Analysis of Theoretical Bounds

Building upon the result in Theorem 6.4, we derive the corresponding asymptotic behavior. First, we observe that the additive term in the bound is subdominant in $T$ and can be ignored asymptotically. Furthermore, we have:

$$\beta(\delta, \boldsymbol{A}_{T_s}) = \mathcal{O}\left(\sqrt{d\log T}\right) \ .$$

Also, by comparing the confidence terms under collaborative and local designs, we obtain:

$$\frac{1}{\sqrt{\rho_{\min}N}} \cdot \frac{\beta(\delta, N\boldsymbol{A}_t)}{\beta(\delta, \boldsymbol{A}_t)} = \mathcal{O}\left(\frac{\sqrt{d}}{\sqrt{\rho_{\min}N}}\right) \ ,$$

which leads to the following asymptotic regret bound:

$$R_{i,0,T}^{\text{cluster}} = \mathcal{O}\left(\frac{\sqrt{d}}{\sqrt{\rho_{\min}N}} \cdot \sqrt{T} \cdot \log\left(\frac{T}{d}\right)\right) \ .$$

Regarding the expected number of misassigned agents, we use the approximation:

$$\sqrt{\det(\boldsymbol{A}_t)} = \mathcal{O}(t) \ ,$$

which implies the asymptotic expression:

$$\mathcal{O}\left(\frac{N}{T^{\gamma^2 R^2 - 1/2}}\right) \ .$$

This analysis, in conjunction with previous results, such as those of Gentile et al. (2014); Li et al. (2019); Ghosh et al. (2022), supports the comparison table.

## P.3. Summary of Clustering Assumptions

We summarize below the main clustering assumptions considered in the literature, as well as those adopted in our work:

- *No assumption:* considered in our paper, except for the setting of Theorem 6.4.

- *Identical bandit parameters within each cluster:* used in our paper under Assumption 6.1.

- *Identical bandit parameters within each cluster, with a minimum separation between clusters:* commonly assumed in prior work such as Gentile et al. (2014); Li et al. (2016); Li & Zhang (2018); Ghosh et al. (2022); Wang et al. (2023); Yang et al. (2024).

- *Approximately similar parameters within clusters, allowing overlapping cluster memberships:* considered in Ban & He (2021).

## Q. Appendix: Implementation details

At each iteration, our *BASS* algorithm calls at most $N - 1$ time the function $\Psi$, which involves to inverse multiple matrices. We propose an efficient implementation involving the use of the Cholesky decomposition, the Sherman-Morrison identity or the eigenvalues decomposition. We report here the implementation details for the main parts of the *BASS* algorithm and their associated complexity.

**Agent separation test**  Recall that the ellipsoid separation test relies on the function defined in Definition 5.1, which involves computing:

$$\forall s \in ]0, 1[ \quad \psi_{i,j}(s) = \frac{\gamma^2}{4} - \sum_{l=1}^{d} \mu_l^2 \frac{s(1-s)}{\beta_i^2 + s(\beta_j^2 \eta_l - \beta_i^2)} \quad ,$$

with $\boldsymbol{\mu} = \boldsymbol{\Phi}^\top \boldsymbol{v}$ such as $\boldsymbol{\eta}$ and $\boldsymbol{\Phi}$ are the eigenvalues and the eigenvectors of the matrix

$$\boldsymbol{Q} = \frac{\beta_i}{\beta_j} \operatorname{chol}(\boldsymbol{A_j})^\top \boldsymbol{A_i}^{-1} \operatorname{chol}(\boldsymbol{A_j}) \quad ,$$

with chol() being the Cholesky decomposition. Moreover, one can derive the first and the second derivatives of $\psi_{i,j}$, and performs the minimization of $\psi_{i,j}$ with the Newton's method. The eigen decomposition features a complexity of order $\mathcal{O}\left(d^3\right)$ and the computation of the first and second derivatives has a complexity of order $O\left(d\right)$. Hence, the complexity of the separation test is of order $\mathcal{O}\left(\tau d + d^3\right)$, with $\tau$ being the number of iterations of the minimization algorithm. However since we use a second order optimization method, in practice only a couple of iterations is needed (*i.e.*, $\tau < 50$), so practically the separation test is of order $\mathcal{O}\left(d^3\right)$.

**Arm selection**  Recall that the arm selection strategy, at iteration $t$, for an agent $i_t$ is defined as $k_t = \arg\max_{k=1,\ldots,K} \widehat{\boldsymbol{\theta}}_{\widehat{\mathcal{N}}_{i_t,t-1}}^\top \boldsymbol{x_k} + \beta_{\boldsymbol{A}^{-1}_{\widehat{\mathcal{N}}_{i_t,t-1}}}(\boldsymbol{x_k})$  with $\beta_{\boldsymbol{A}^{-1}_{\widehat{\mathcal{N}}_{i_t,t-1}}}(\boldsymbol{x}) = \alpha\sqrt{\boldsymbol{x}^\top \boldsymbol{A}^{-1}_{\widehat{\mathcal{N}}_{i_t,t-1}} \boldsymbol{x} \log(t)}$ which features a complexity of $\mathcal{O}\left(Kd^2\right)$.

Following the different implementations detailed here, the overall complexity of the *BASS* algorithm is of order $\mathcal{O}\left(T(N + Kd^2 + (N-1)d^3)\right)$. We notice that the *BASS* algorithm as a linear complexity toward the number of agents $N$, the number of arms $K$ and the number of iterations $T$. However, it features a cubic dependence toward the dimension $d$ of the problem.

## R. Appendix: Experimental details

**Description of the concurrent algorithms**  For the benchmarks, we select four other concurrent algorithms: *DynUCB*, *CLUB*, *SCLUB*, *CMLB*. First, *DynUCB* is a simple yet efficient approach to perform agent clustering in a linear bandit setting. Proposed in Nguyen & Lauw (2014), it relies on the $k$-means algorithms MacQueen (1967) to update the clusters at each iteration. This algorithm needs to set the number of estimated clusters. For this experiment, we set it to the true number of clusters. To our knowledge, no other method proposes this $k$-means-based approach for this class of problems, which makes it of particular interest. Second, *CLUB* and its improved version *SCLUB* are likely the most benchmarked algorithms for the clustering linear bandit problem and can be considered the reference for it. *CLUB* was introduced in Gentile et al. (2014) and *SCLUB* resp. in Li et al. (2019), both propose to determine whether two agents belong to the same cluster by performing a test on the $\ell_2$-norm of the difference of their bandit parameters. The improved version *SCLUB* provides a more flexible cluster estimation by iteratively splitting and merging the clusters. Finally, we consider *CMLB* form Ghosh et al. (2022) which is a more recent approach closely related to *CLUB* using the same method to determine whether two agents belong to the same cluster. However, this approach controls the minimum number of agents in a cluster. Note that these last three algorithms need to set a scalar to scale the clustering threshold that determines whether this difference is significant.

**Synthetic experiment settings**  In Section 7, in the synthetic experiment, we consider $M = 3$ clusters of the same size with bandit parameter $(\boldsymbol{\theta}_m)_{m \in \{1,\ldots,M\}}$ defined as: $\forall 1 \leq q \leq \lceil \frac{M}{2} \rceil \, \boldsymbol{\theta}_{2q-1} = \boldsymbol{e}_q$, with $(\boldsymbol{e}_i)_i$ being the canonical basis and $\boldsymbol{\theta}_{2q}$ having its $q$-th entry being $\cos\omega$, its $(q+1)$-th entry being $\sin\omega$ and all the other entries being 0, with $\omega \in \{\pi/16, 7\pi/16\}$. Indeed, these configurations are of particular interest because they correspond to the case where the bandit parameters are

positively correlated and almost orthogonal. With $\omega = \pi/16$, the bandit parameters $(\boldsymbol{\theta}_m)_{m\in\{1,...,M\}}$ are similar. Thus, assigning the wrong cluster will not overly perturb the agent, even though discriminating clusters will be tougher. The second scenario with $\omega = 7\pi/16$ the two bandit parameters are almost orthogonal which makes them easier to separate one another and thus identify the cluster. However, the two models are very different thus if an agent is assigned to the wrong cluster, he will most likely perform poorly once he will gather the observations of the other agents. For the *CLUB*, *SCLUB* and *CMLB* algorithms we test two levels of clustering exploration $\gamma \in \{\frac{\Delta_{\theta^*}^{\min}}{4}, \frac{\Delta_{\theta^*}^{\min}}{2}\}$, the result is shown in the 'dashed' and 'solid' lines. For *DynUCB*, we explore $M = 5$ and $M = 20$ for the $k$-means algorithms MacQueen (1967) with the same visual coding for the two cases. For the *BASS* algorithm, we test two values for $\delta \in \{0.1, 0.9\}$ equivalently, with the same choice of result visualization. We keep the same color code for the rest of the experiment: *DynUCB* in pink, *CLUB* in blue, *SCLUB* in brown, *CMLB* in red and our approach *BASS* in orange, *Ind* in light gray and when available *Oracle* in dark gray. We choose the $\alpha$-*LinUCB* agent policy for our algorithm and line-search the UCB parameter $\alpha$ within $[0.1, 3.0]$.

**Clustering score computation**     To quantify the degree of agreement between the estimated clusters/neighborhoods and the ground truth, we compute the $F_1$-score on the graph $\mathcal{G}_t$ w.r.t. the true cluster graph $\mathcal{G}$. For the *DynUCB* we define $\mathcal{G}_t = (V, E_t)$ such as $E_t = \{i, j | 1_{\{\text{kmeans}-\text{label}(i)=\text{kmeans}-\text{label}(j)\}}\}$. The clustering score will output a value within $[0, 1]$, where 1 is a perfect cluster match.

**Description of the datasets and experiment settings**     Recall that in Section 7, we perform experiments on two real datasets. We detail here their characteristics. First we select the *MovieLens* dataset[5] Harper & Konstan (2015): which is a dataset that proposes the rating (from 1 to 5) of 62,000 movies by 162,000 users for a total of 25 million ratings. Each movie is represented by features describing its genre and historical records. Since the users did not rate all the movies systematically, we fill the missing values with a matrix-factorization approach Rendle & Schmidt-Thieme (2008). From this complete dataset, we are able, for a given user—agent—and a given movie—arm—, to fetch the corresponding reward.

Second, we consider, *Yahoo!* dataset[6] Chapelle & Chang (2010): which is a dataset of recommendations made from queries on the Yahoo! search engine. For each query, documents are received, represented by sparse features vectors of dimension 500 and their corresponding reward (from 1 to 4) which measures how relevant the returned documents were. We consider the queries as—agents—and the document as the—arms—, to reduce their cardinality, we perform the $k$-means algorithm and retained only $K = 30$ arms. As with the previous dataset, we complete the preprocessing by filling the missing rewards with the same matrix-factorization approach to obtain, for each user, a reward for every arm.

We consider the same setting as the synthetic experiment: we select $N = 100$ agents and reduce the arm dimension to $d = 10$ by taking the $d$ first dimensions of the native data space. Since the randomness of our preprocessing is the user's draft to be retained on the $N$ agents, we run the experiment 50 times to average across runs.

## S. Appendix: Additional details

### S.1. Appendix: Additional synthetic experiment results

To complete the analysis with the synthetic data scenario considered in Section 7, we report here the evolution of the cumulative regret value in the case of $M = 3$ and $M = 6$. We consider the same experimental setting as in the synthetic data experiment of Section 7.

---

[5]Available at https://grouplens.org/datasets/movielens/
[6]Available at https://webscope.sandbox.yahoo.com/catalog.php?datatype=c

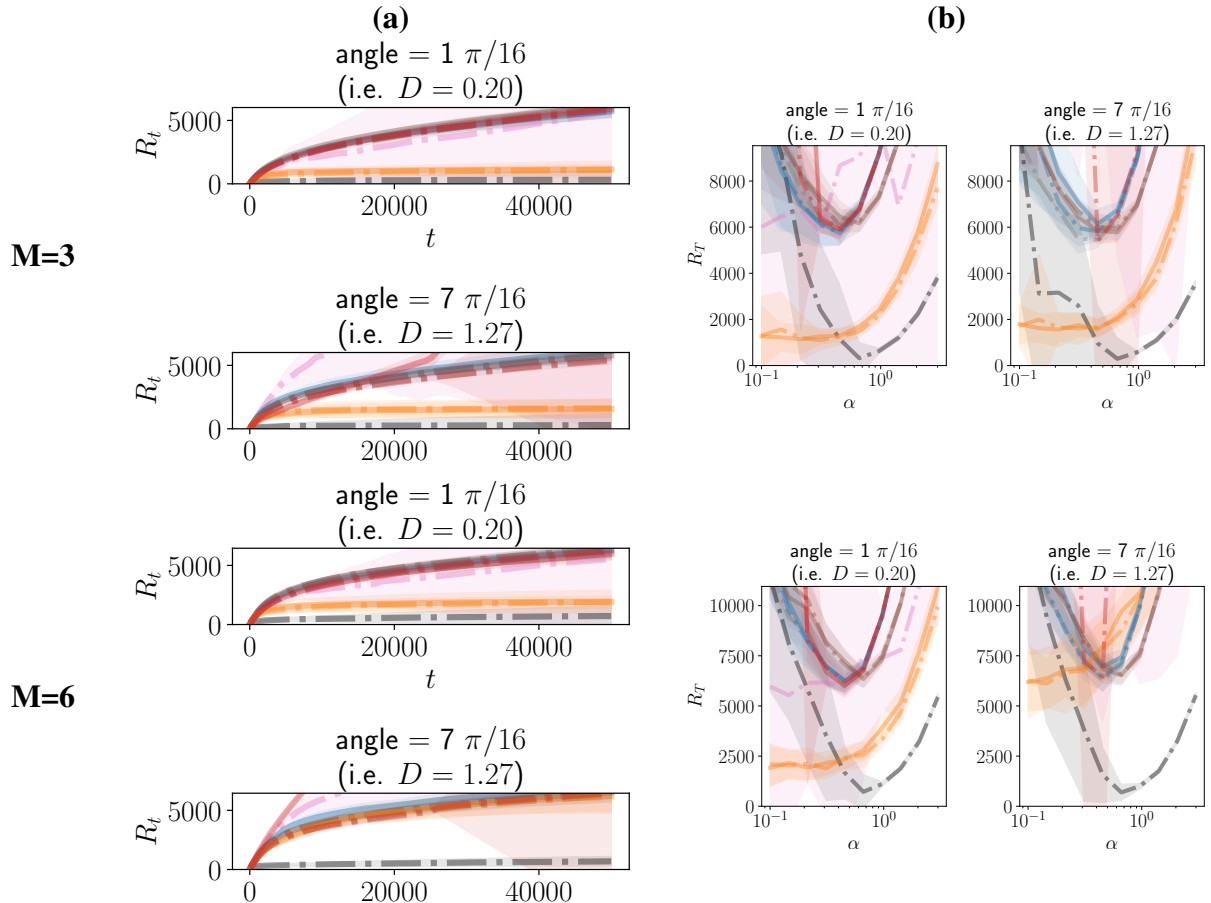

*Figure 6.* **(a)** Comparison of the averaged, across runs, cumulative regret evolution $(R_t)_t$ for the synthetic environments considered, $M = 3$ on the top and $M = 6$ on the bottom. **(b)** Comparison of the averaged, across runs, evolution of the cumulative regret last value $R_T$ w.r.t. the UCB parameter $\alpha$ for the different synthetic environments considered, with the same color code as previously.

In Figure 6 **a** we display the cumulative regret evolution $(R_t)_t$, $M = 3$ on the top and $M = 6$ on the bottom and in Figure 6 **b**, we display the cumulative regret last value, $R_T$, evolution w.r.t the UCB parameter $\alpha$, $M = 3$ on the top and $M = 6$ on the bottom, both for all the concurrent and baseline methods. The overall performance comparison between the algorithms stays coherent with all the previous experiments and confirms the robustness and the good behavior of our algorithm. Interestingly, for *BASS*, we systematically observe an inflection of the curve to a quasi no-regret plateau.

### S.2. Appendix: Additional real data experiment results

To complete the analysis with the real data scenario considered in the paper, we report here the evolution of the cumulative regret last value w.r.t. the UCB parameter.

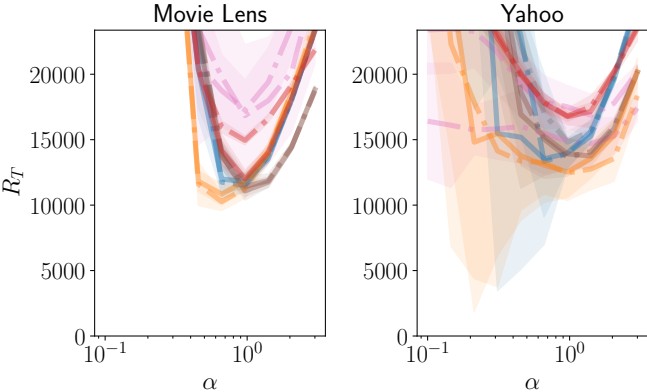

*Figure 7.* Comparison of the averaged, across runs, evolution of the cumulative regret last value $R_T$ w.r.t. the UCB parameter $\alpha$ for the different synthetic environments considered, with the same color code as previously.

In Figure 7, we display the cumulative regret last value, $R_T$, evolution w.r.t. the UCB parameter $\alpha$. We notice that again *BASS* performs better than the other algorithms, it is more pronounced in the case of the *Yahoo* dataset.

### S.3. Appendix: Additional benchmark experiment

Additionally, we consider a simpler benchmark to underline the robustness of the results depicted in the paper. In this benchmark, we consider the same experimental setting as in the synthetic data experiment of Section 7, but in this case, we randomly draw the bandit parameters and the arms from a standard Gaussian distribution and consider $M = 3$. We consider three levels of noise with $\sigma \in \{0.5, 1.0, 2.0\}$

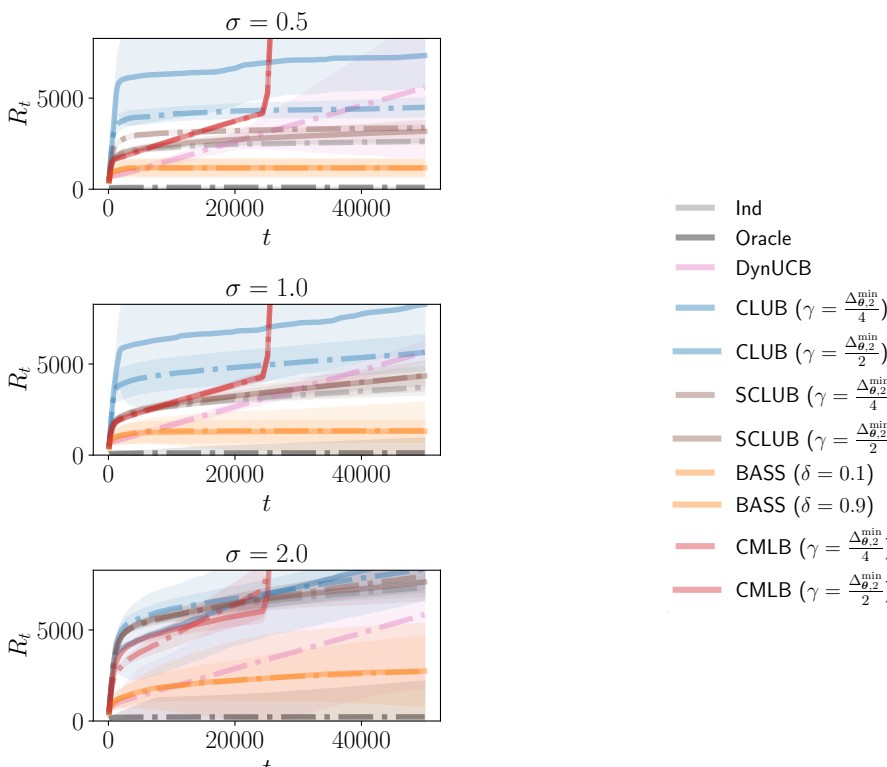

*Figure 8.* Comparison of the averaged, across runs, cumulative regret evolution $(R_t)_t$ for the different synthetic environments considered with $M = 3$.

In Figure 8, we display the cumulative regret evolution $(R_t)_t$ for all the concurrent and baseline methods. We notice the

good performance of the *BASS* algorithm and interestingly in the case where $\sigma = 2.0$, we notice that the difference of performance compare to the other increase, which emphasize how robust to the noise level our algorithm is.

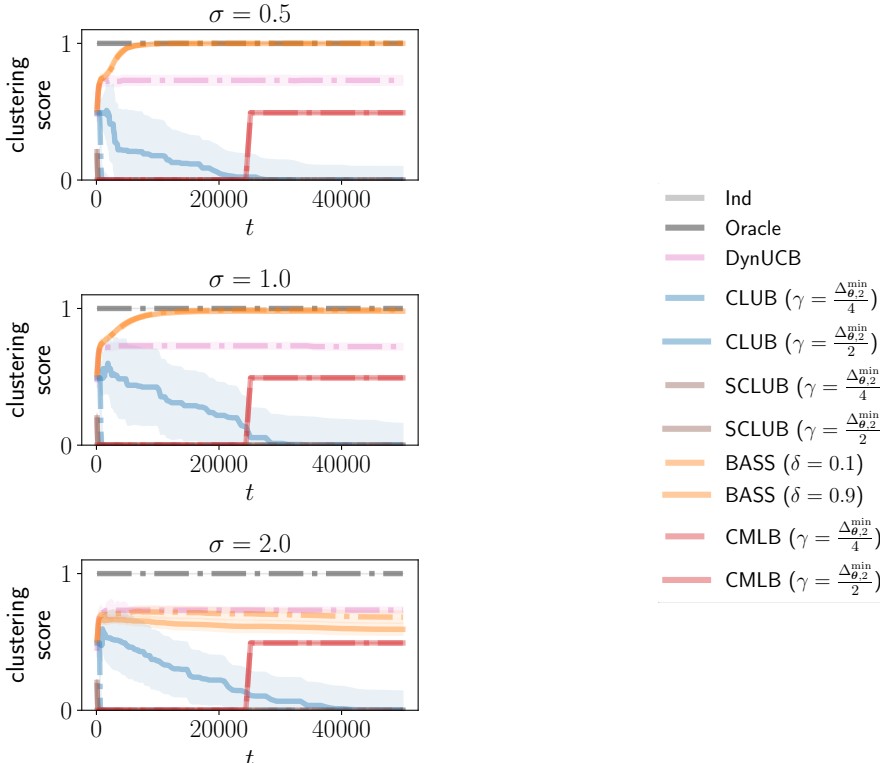

*Figure 9.* Comparison of the averaged, across runs, clustering score evolution for the different synthetic environments considered with $M = 3$.

In Figure 9, we display the clustering score evolution for all the clustering algorithms. We quantify the clustering estimation quality as previously and notice that our approach achieves again the best performance. Indeed, we observe that most of the concurrent algorithms did not manage recover any clusters.

