# OpenReview forum: "Adaptive Sample Sharing for Multi Agent Linear Bandits"
_ICML.cc/2025/Conference — ICML 2025 poster_

### Official Review · Reviewer_R3aR · 2025-03-03

**Overall Recommendation:** 3

**Summary:**

The paper studies an adaptive sample sharing problem for multi-agent linear bandits, where agents' true parameters may be different. The authors propose a separation test, which detects the stopping time for beneficial collaboration. The authors provide both the separation time upper-bound and cumulative pseudo-regret upper bound. The authors also compared the pseudo regret with existing methods. Finally, the authors empirically conduct experiments using both synthetic data and real-world data.

**Claims And Evidence:**

I have several confusions when reading the paper.

1. Can you elaborate on the bias-variance tradeoff in Figure 1? Why is the orange ellipsoid inside the blue ellipsoid?

2. I'm confused of the connection between Table 1 and Theorem 6.4? How do you get the $\sqrt{T}\log(T/d)$ upper bound from Theorem 6.4?

**Essential References Not Discussed:**

I am not very familiar with the collaborative learning literature in linear bandits so I'm unsure whether essential references are missed.

**Experimental Designs Or Analyses:**

I do not find big issues in experiments. This paper is mostly a theoretical work.

**Methods And Evaluation Criteria:**

The paper studies the separation time and cumulative regret of the proposed algorithm, which looks reasonable to me.

**Other Comments Or Suggestions:**

I'm curious why the authors define "pseudo regret" instead of "regret". The definition looks the same as the standard regret.

**Other Strengths And Weaknesses:**

The paper provides detailed upper bound analysis on pseudo regrets, the number of misaligned agents, and separation time. They also provides the lower bound on separation time. However, the lower bound for pseudo regrets is missed. How close is your algorithm's upper bound to the lower bound?

**Questions For Authors:**

In Figure 1, how do you compare BASS with CLUB and SCLUB? The constant term of SCLUB and CLUB depend on $M$, and your algorithm's upper bound has different parameters. It's unclear why BASS has tighter upper bound.

**Relation To Broader Scientific Literature:**

This research is related to the broader multi-agent collaborative learning, where multiple agents collaboratively share data for learning purposes.

**Theoretical Claims:**

I feel the theoretical parts are not well presented. More specifically in Theorem 6.4, what is $v(\delta, T)$? How do you quantify $n_i^e(t)$? The authors should have a remark to discuss the quantity of the bound. Currently, the upper bound is not very clear to me.

---

> ### Author Rebuttal · Authors · 2025-03-31
>
> We would like to thank the reviewer for their positive and constructive comments. Below, we address your concerns and provide detailed answers to your questions.
>
> 1/ The orange ellipsoid (representing the collaborative estimation) corresponds to a reduced estimation variance, as it yields a smaller Mahalanobis radius of the confidence ellipsoid—note that the smaller ellipsoid might not be contained in the bigger one, we will provide a more general illustration to underline that. However, notice that the center of the orange ellipsoid differs from that of the blue ellipsoid due to the introduced bias, illustrating the bias (OLS estimate deviation) - variance (confidence ellipsoid radius reduction) tradeoff.
>
> 2/ We will add a section in the appendix to properly detail the steps to obtain Table 1 from Theorem 6.4 and Lemma 6.1. The proof is similar to CLUB and SCLUB, and the improvements stem from both the Mahalanobis separation criterion and a refined analysis from Abbasi-Yadkori et al 2011.
>
> 3/ The definition of $\nu$ remains unchanged across all theorems: it is the regret upper bound without sample sharing. Considering the upper bound of Theorem 6.4, the first term corresponds to the regret minimization improvement due to the variance reduction (represented by $\mu$) and the second term corresponds to the cost of sharing (i.e., the shared OLS estimate bias, due to misassigned agents). This second term features an interplay between the determinant and the exponential term in $n_i^e(t)$, which overall results in a $\mathcal{O}(\frac{4L}{\delta \rho_{\min}\sqrt{T}^{\gamma^2R^2 - 1}})$ regret. This leads to an overall behavior of the upper bound in $\mathcal{O}(\sqrt{T} \log (T/d))$ as depicted in Table 1.
>
> 4/ Thank you for pointing out the inconsistency in the appendix content table; we will revise the naming of the appendix sections accordingly.
>
> 5/ To derive a lower bound on the cumulative pseudo-regret, we consider the optimal case of $N$ collaborative agents sharing the same bandit parameters, forming a single—known—cluster. This leads to: $\frac{1}{N}\mu(\delta, T) \leq R_{i,0,T}^{\mathrm{cluster}}$. We will further compare the additional terms stemming from the clustering estimation in the upper bound of Theorem 6.4 to this lower bound to underline the cost of clustering.
>
> 6/ As defined by in [1], pseudo-regret uses expected rewards, while regret uses realized rewards; for our paper we use Bubeck’s definition.
>
> 7/ Thank you for pointing out the typo in Table 1: the proper constant term should be $d\sqrt{\frac{M}{N}}$ for CLUB and SCLUB which is equivalent to $\frac{d}{\sqrt{\rho_{\min}N}}$ for balanced cluster sizes. As mentioned in 2/, the differences ($\sqrt{d}$ factor and $T/d$ in the logarithm)  are due to the use of a Mahalanobis metric combined with a refined analysis (i.e., applying the Elliptical Potential lemma on the design matrix $\mathbf{A}_i$).
>
> [1] Bubeck, S., & Cesa-Bianchi, N. (2012). Regret analysis of stochastic and nonstochastic multi-armed bandit problems. Foundations and Trends® in Machine Learning, 5(1), 1-122.
>
> Additional comments:
>
> If you have any additional questions about the paper and the theoretical and empirical analysis, we would be happy to provide further clarification.
>
> Please let us know if you need any additional information, clarification or modification that we could provide for you to improve your score.

---

> > ### Comment · Reviewer_R3aR · 2025-04-03
> >
> > Thank you for your detailed responses. I decide to increase my score to 3.

---

### Official Review · Reviewer_vhoL · 2025-03-04

**Overall Recommendation:** 4

**Summary:**

This paper considers the problem of multi-agent linear bandits in a collaborative setting where the aim is to maximise the cumulative reward across all agents. In this problem each agent has the same (static) set of arms but has a different parameter. The idea is if the parameters of two agents are close enough together then pooling their results to create a single parameter estimator is better than learning independently.

Whereas previous works assume that the agents form clusters in which the intra-cluster parameter distances are within a certain threshold, this work has no such assumptions. They also show that in the clustering case their algorithm theoretically outperforms existing algorithms for that case. The algorithm is also relatively efficient.

They also give the results of experiments, both on real and synthetic data, showing that their algorithm outperforms others.

**Claims And Evidence:**

I have not read the proofs so can’t confirm

**Essential References Not Discussed:**

Unknown

**Experimental Designs Or Analyses:**

I did not check

**Methods And Evaluation Criteria:**

Yes

**Other Comments Or Suggestions:**

I very much like the way the paper is presented - building up from the fundamental principles of two agents sharing observations and the single agent stochastic linear bandit. However, in my subjective opinion I believe that it is best to place the full theoretical result as soon as possible in the work (which would be before these sections).

**Other Strengths And Weaknesses:**

I feel the result is strong. But also, as far as I am aware from interacting with ChatGPT, the original clustering algorithm of Claudio Gentile et. Al. treats all agents in a cluster the same when it has learnt it. If significant time passes the strategy of playing each agent the same in a given cluster is non optimal and it is better to learn them independently, whilst the algorithm of this paper appears to eventually learn them independently, which seems to be another strength (although I guess other clustering algorithms have been proposed that do this).

**Questions For Authors:**

None

**Relation To Broader Scientific Literature:**

There has been much work on collaborative multi-agent linear bandits. This paper gives the first algorithm that makes no assumptions on the distances between the parameters of the agents. Most other works assume a clustering assumption, and this work also shows that the same algorithm has a state of the art result there as well (albeit only a slight improvement on the previous state of the art). These two facts combined make this work a significant contribution to the field.

**Theoretical Claims:**

I have not read the proofs

---

> ### Author Rebuttal · Authors · 2025-03-31
>
> We would like to thank the reviewer for their very positive and insightful comments. Thank you very much for your kind appreciation of the paper's results.
>
> Thank you for the suggestion regarding the structure of the paper; we will revise it, as we agree it improves the overall logical flow.
>
> Additional comments:
>
> If you have any additional questions about the paper and the theoretical and empirical analysis, we would be happy to provide further clarification.

---

### Official Review · Reviewer_R5Cg · 2025-03-12

**Overall Recommendation:** 4

**Summary:**

This paper studies the collaboration of multiple heterogeneous agents in addressing a linear bandit problem. It proposes an adaptive sample approach to dynamically determine whether agent pairs should cooperate (utilize observations). Based on the technique, the paper proposes the BASS algorithm to address the multi-agent linear bandit problem with a tighter theoretical regret bound and better empirical performance.


#### After rebuttal

This is a nice piece of work. The reviewer would like to see the final version with all the comments addressed accordingly.

**Claims And Evidence:**

Yes

**Essential References Not Discussed:**

No

**Experimental Designs Or Analyses:**

Yes, the simulations look reasonable to the reviewer.

**Methods And Evaluation Criteria:**

Yes

**Other Comments Or Suggestions:**

In Section 5:
- The presentation logic also makes the reviewer a bit confused. At the end, the action sequence, or $\bm A_i$, are different across agents. Why the section starts from the general case in Eqs. (3) and (4), and then go to the homogeneous sequence case in Lemma 5.2, and then go back to general case in Definition 5.1? Is there a challenge of extending from Lemma 5.2 to Definition 5.1?
- In the equation of definition 5.1, the reviewer is a little bit lost: what is $s$ in the definition, and where $t$ as an input to $\Psi$ appears in the equation?

In Section 6:
- In Algorithm 1’s Line 3, how the neighbors $\hat{\mathcal{N}}$ are determined by the updated graph $\mathcal{G}_t$ in Line 9, or by the environment?

**Other Strengths And Weaknesses:**

Overall, the reviewer believes the paper makes a fair contribution, and thinks the approach can be extended in broader areas, like other heterogeneous multi-agent bandits setting, and MARL as well.

### Strengths:
- New approach and better results: the adaptive proof discussed in Sections 3.2 and 5 is new to the reviewer, and the BASS algorithm based on the technique is also novel in the literature with slightly tighter regret bounds.

- Clear writing: the authors did a good job in presenting their work, with toy two-agent as an example, and then extended to multi-agent, with synchronous action as a start, and then generalized to heterogenous actions. The illustrative figures (Figures 1 and 2) also help the reviewer better appreciate the fundamental idea behind the approach.

### Weakness:
- Some sections can be merged; for example, Section 4 should be in the Preliminaries, and Section 7 should just be put as a remark for Theorem 6.4 instead of a full section.
- Some presentation/writing needs further clarifications, see "Other Comments" below.

**Questions For Authors:**

See above

**Relation To Broader Scientific Literature:**

The idea proposed in this paper may initiate a broader impact in the community of sequential decision-making and reinforcement learning.

**Theoretical Claims:**

The reviewer does not check the proof in detail, but the reviewer goes through the detail explanation in the main text and believes the intuition before the theoretical results of this paper make sense.

---

> ### Author Rebuttal · Authors · 2025-03-31
>
> We would like to thank the reviewer for their positive and constructive comments. Below, we address your concerns and provide detailed answers to your questions.
>
> 1/ Thank you for the remark; we will revise the structure of our paper, as we agree it improves the overall logical flow.
>
> 2/ There is indeed a challenge in analyzing the effects of asynchronous pulling. It leads to differently shaped ellipsoids and requires a rescaling term in the form of $\mathbf{A}_i^{-1} \mathbf{A}_j$, which makes the proof much more difficult. Therefore, we restrict our analysis to the synchronous setting. In Definition 5.1, we use the general case because this definition is used in the practical implementation of the algorithm, which allows for empirical evaluation of our method in the asynchronous setting.
>
> 3/ Thank you for pointing out this typo: the correct definition consider the minimum w.r.t $s$ and compare the later to 0:
> $$
> \Psi(i, j, t) = \mathbf{1}\left \\{ \min_{s \in ]0, 1[} \frac{\gamma^2}{4} - \sum_{l=1}^d \mu_{tl}^2 \frac{s (1 - s)}{\beta_{i,t}^2 + s (\beta_{j,t}^2\eta_{tl} - \beta_{i,t}^2)} < 0 \right\\} \enspace.
> $$
> We will correct this and update the corresponding subscripts.
>
> 4/ The graph $\mathcal{G}_t$ is updated and stored in a central server that first requests the design matrix and the regressand of each agent in the neighborhood of agent $i$. Then, send collaborative OLS variables to agent $i$ to pull an arm. Hence the neighborhood is determined by a central server to manage the sampling sharing logic within the agents network. Note that our work focuses on sample efficiency rather than communication; however, sample sharing for a general decentralized setting is a promising line of future work.
>
> Additional comments:
>
> If you have any additional questions about the paper and the theoretical and empirical analysis, we would be happy to provide further clarification.

---

> > ### Comment · Reviewer_R5Cg · 2025-04-01
> >
> > The reviewer thanks the author for the response. The reviewer will hold their positive evaluation on the paper.

---

### Official Review · Reviewer_VM4K · 2025-03-20

**Overall Recommendation:** 3

**Summary:**

This paper considers a multi-agent linear bandit problem in which each agent seeks to estimate its own linear parameter (so that they can minimize regret) while all agents select arms from a shared set. In this setting, agents are allowed to share reward observations with other agents to reduce the uncertainty of parameter estimates at the cost of increasing the bias of the estimates. To construct proper collaboration sets for agents, the authors leverage the idea of overlapping ellipsoid test, proposing an OFUL-based algorithm with Mahalanobis distance-based agent similarity criterion. This paper provides theoretical guarantees for cumulative pseud-regret, agent separation time (length of communication), and the expected number of misassigned agents. The proposed algorithm is also numerically evaluated in both synthetic and real-world datasets.

**Claims And Evidence:**

This paper claims three key contributions: no parameter assumption, anisotropic approach, and analysis. However, as one of the main contributions, the algorithm description in Section 6.1 appears too brief. Could the authors provide a more detailed explanation to enhance clarity?

**Essential References Not Discussed:**

I am not aware of essential references that were not discussed.

**Experimental Designs Or Analyses:**

Could the authors provide comments on the results presented in Table 3? In particular, could they offer an explanation or intuition for why almost all benchmark algorithms received zero scores? Additionally, could the authors discuss whether the approaches proposed in this paper could be incorporated into these algorithms?

**Methods And Evaluation Criteria:**

This paper employs appropriate metrics, such as cumulative pseudo-regret and clustering scores, to evaluate performance. However, while the number of communications (i.e., agent separation time) appears to be considered in the theoretical analysis, it is not reported in the numerical study. Could the authors clarify this or provide the corresponding simulation results?

**Other Comments Or Suggestions:**

N/A

**Other Strengths And Weaknesses:**

Strength:
- This paper proposes an interesting algorithm for multi-agent linear bandit and provides extensive theoretical analysis.

Weakness:
- Some of the results are not presented clearly.

**Questions For Authors:**

- I am trying to better understand the claim that, unlike "most" existing approaches, this work does not rely on any assumptions about the structure of the bandit parameters. Do works that assume an agent cluster structure (e.g., Gentile et al. 2014, Ban & He 2021, Wang et al. 2023) impose assumptions on the parameters? Could the authors clarify this point?

- I am puzzled by the discussion following Lemma 5.1, which states that as we consider synchronous pulling, we have $ {\bf A}\_{i, t} = {\bf A}\_{i, t} = {\bf A}\_t$. My understanding is that even if agents $i$ and $j$ are synchronous, if they pull different arms, then ${\bf A}\_{i, t} \neq {\bf A}\_{j, t}$. Could the authors clarify this point? Please let me know if I have misunderstood anything.

**Relation To Broader Scientific Literature:**

This work may bring attention to the potential of incorporating Mahalanobis distance as a similarity measure in the multi-agent regret minimization research community.

**Theoretical Claims:**

- In Theorem 6.2, the term $ \beta(\delta, {\bf A}_{T_s}) $ is not explicitly specified.
- In Theorems 6.3 and 6.4, the terms $ \nu(\delta, T) $ are not explicitly specified. Could the authors clarify whether they take the same form as in Theorem 6.2?
- I am seeking a better understanding of the general regret upper bound for all agents presented in Theorem 6.3. Could the authors summarize the assumption-free bounds in a similar manner to Table 1 for clarity?

Providing these clarifications in the paper would enhance readability.

---

> ### Author Rebuttal · Authors · 2025-03-31
>
> We would like to thank the reviewer for their positive and constructive comments. Below, we address your concerns and provide detailed answers to your questions.
>
> 1/ We will add a specific section in the appendix, featuring a flow-chart, where we will better expand the description of the algorithm major steps: a/ the samples sharing (L 2-4), b/ the UCB pulling (L 5-6), c/ the local variables update (L 7) and d/ the graph $\mathcal{G}_t$ update (L 9).
>
> 2/ The focus of this paper is to analyze the effect of sample sharing on regret minimization in the general case—i.e., without any assumption on the agent clustering structure. To that end, we ignore considerations such as communication cost. However, for completeness, we will add a section in the appendix presenting the empirical communication cost in the federated setting (using a central server), along with a theoretical upper bound in the case where the clustering structure is assumed (since in that setting, the number of agents per cluster can be quantified).
>
> 3/ The term $\beta(\delta, \mathbf{A}_t)$ is stated in Theorem 4.1 and corresponds to the confidence ellipsoid radius of the OLS estimate. We will restate and clarify the definition of $\beta(\delta, \mathbf{A}_t)$ in Theorems 5.1, 6.1, 6.2, 6.3, and 6.4, and underline that the definition of $\nu$ remains unchanged across these theorems.
>
> 4/ A particularity of the sample sharing problem is that it unfolds in two distinct phases: one where sharing is beneficial, and another where independent parameter estimation is preferable. Theorems 6.2 and 6.3 present the cumulative regret at the transition point between these phases, highlighting the gain from sharing before switching to an independent regime. Table 1 reports the asymptotic regret in the independent setting, after collaboration has ceased. Thus, the results of Theorems 6.2 and 6.3 cannot be presented in the same form. However, the upper bound can be summarized as a tradeoff between the acceleration of the first term (in $\mu$), compared to the independent regret ($\nu$), and the bias from data sharing, captured in the second term.
>
> 5/ The other algorithms perform poorly in terms of clustering. Rather than accurately recovering the cluster structure, other approaches focus on regret minimization, either by design or by regret-oriented hyperparameter optimization. Figure 8 in Appendix R.3 shows the progressive degradation of the clustering score. Notably, no existing work in the literature provides theoretical guarantees or empirical validation regarding the quality of the clustering.
>
> 6/ Beyond the analysis of sample sharing itself, the most significant aspect to revise is the use of the Euclidean distance criterion (as in Gentile et al. (2014); Ban & He (2021); Wang et al. (2023)), which is effectively equivalent to the test proposed by Gilitschenski et al. (2012), where the ellipsoids considered are $\mathcal{E}(\hat{\mathbf{\theta}}_i, \lambda^{\min}(\mathbf{A}_t)\mathbf{I}, \tilde{\beta})$ and $\mathcal{E}(\hat{\mathbf{\theta}}_j, \lambda^{\min}(\mathbf{A}_t)\mathbf{I}, \tilde{\beta})$  thereby ignoring most of the arm-pulling history.
>
> 7/ We will add a summary table in the appendix outlining the different clustering assumptions considered in the literature and mentioned in our related work:
> * no assumption: our paper, except in “Regret analysis with clustered parameters”.
> * same bandit parameter within a cluster: our paper with Assumption 6.1.
> * same bandit parameter within a cluster and a minimum distance between clusters: Gentile et al. 2014, Li et al. 2016, Li & Zhang, 2018, Wang et al. 2023, Yang et al. 2024.
> * close bandit parameters within a cluster and overlapping clusters definition:  Ban & He 2021.
>
> Note that our primary intention is to derive a principle rule to share samples between agents without introducing assumptions on their distribution. We then extend our results to the clustered setting to provide a fair comparison to existing work.
>
> 8/ In the synchronous setting, all agents in a cluster select an arm based on the same shared history. Since UCB is a deterministic rule, two agents using the same history will select the same arm. Therefore, their design matrices will also be identical. We will add an additional section in the appendix to better explain this aspect of our setting.
>
> Additional comments:
>
> If you have any additional questions about the paper and the theoretical and empirical analysis, we would be happy to provide further clarification.
>
> Please let us know if you need any additional information, clarification or modification that we could provide for you to improve your score.

---

> > ### Comment · Reviewer_VM4K · 2025-04-02
> >
> > I greatly appreciate the authors' clarification in response to my questions, as well as their plan to incorporate additional material into the paper. I continue to maintain my positive recommendation for this work.

---

### Decision · Program_Chairs · 2025-05-01

**Decision:**

Accept (poster)

**Comment:**

I appreciate the work as collaborative learning in bandits is an important and practical topic and the results obtained in this paper are solid. Two of the four reviewers are on the fence and remain so after the rebuttal. While the overall evaluation is a bit on the borderline, I'm leaning towards accepting the paper and concur more with the two reviewers that are more positive.